# Anionic ring-opening polymerization of functional epoxide monomers in the solid state

Jihye Park [1], Ahyun Kim [1] & Byeong-Su Kim [1] ✉

Despite recent advancements in mechanochemical polymerization, understanding the unique mechanochemical reactivity during the ball milling polymerization process still requires extensive investigations. Herein, solid-state anionic ring-opening polymerization is used to synthesize polyethers from various functional epoxide monomers. The critical parameters of the monomers are investigated to elucidate the unique reactivity of ball milling polymerization. The controllable syntheses of the desired polyethers are characterized via NMR, GPC, and MALDI-ToF analyses. Interestingly, bulky monomers exhibit faster conversions in the solid-state in clear contrast to that observed for solution polymerization. Particularly, a close linear correlation is observed between the conversion of the ball milling polymerization and melting point of the functional epoxide monomers, indicating melting point as a critical predictor of mechanochemical polymerization reactivity. This study provides insights into the efficient design and understanding of mechanochemical polymerization.

Mechanochemical reactions are chemical processes triggered by mechanical energy, such as grinding, shearing, and collisional forces[1]. To date, many researchers have conducted various mechanochemical studies, such as mechanochemical syntheses ranging from organic synthesis and coupling[2,3] to polymerization[4], and the use of mechanoresponsive moieties (*i.e.*, mechanophores)[5]. To understand and control mechanochemical reactions, particularly in the case of ball milling reactions, various parameters influencing the reactivity have been investigated. These parameters typically include the type of milling materials, size and number of balls, period and frequency of milling, grinding auxiliary, and liquid used to assist grinding[4]. Owing to its inherent advantages that can overcome solubility issues and provide access to reactivity and selectivity, the mechanochemical approach has recently expanded to diverse synthetic fields[6–8].

An early example of ball milling polymerization is Gilch polymerization of poly(phenylene vinylene)[9] by Swager et al. Subsequently, Borchardt group performed polycondensation of dialdehydes and diamines[10] and further realized Suzuki polycondensation to synthesize linear and hyperbranched polyphenylenes[11]. Independently, Kim et al. described ring-opening polymerization of lactide[12,13] and trimethylene

carbonate[14] as well as ring-opening metathesis polymerization of norbornene-based monomers and copolymerization of immiscible monomers[15]. More recently, copolymerization of polyurethane with various diamines and diols has been achieved[16], and controlled polymerization of solid 2-vinylnaphthalene has been performed via atom transfer radical polymerization[17].

Despite the considerable advancements in mechanochemical polymerization, the influence of the structures and properties of monomers on the polymerization reactivity remains unclear. Meanwhile, Peterson and Choi have focused on identifying the parameters influencing mechanochemical degradation of various types of polymers, such as linear, bottlebrush, and dendronized polymers, in the ball milling process[18,19]. They have found that milling frequency has the highest impact on the degradation rates of polystyrene and poly(methyl methacrylate) and that the glass transition temperature ($T_g$) of these polymers is also influential.

Herein, we aim to provide a systematic approach toward elucidating the relationship between the mechanochemical polymerization reactivity and properties of monomers. Polyether was specifically selected as the platform for this study because of its negligible chain

---

[1]Department of Chemistry, Yonsei University, Seoul 03722, Republic of Korea. ✉e-mail: bskim19@yonsei.ac.kr

scission during the mechanochemical reaction due to its chain flexibility[7,18] and broad potential applications in biological, cosmetic, automotive, and electrical fields[20]. Despite their versatility and suitability in the ball milling process, the mechanochemical polymerization reactions of polyethers have not yet been reported. In this context, herein, a library of functional epoxide monomers with diverse physical properties was developed for controlled mechanochemical anionic ring-opening polymerization (AROP) via the ball milling process. Specifically, five functional epoxide monomers with varying physical properties were prepared, including 4-methoxyphenyl glycidyl ether (MPG; $R_1$), 3,5-dimethoxyphenyl glycidyl ether (DPG; $R_2$), biphenyl glycidyl ether (BPG; $R_3$), trityl glycidyl ether (TGE; $R_4$), and (s)-trityl glycidyl ether ((s)-TGE; $R_5$) (Fig. 1).

The controllable syntheses of the desired polyethers were characterized using a series of NMR, GPC, and MALDI-ToF analyses. Surprisingly, a clear opposite trend in reactivity was observed between solid-state and solution polymerizations. Notably, the parameter for intermolecular interactions (e.g., melting point) of the functional epoxide monomers was the most critical predictor of ball milling polymerization reactivity.

## Results

### Design and synthesis of the functional epoxide monomers

We initially designed an array of functional epoxide monomers to perform solid-state mechanochemical polymerization of polyethers. The five representative solid-state monomers of MPG ($R_1$), DPG ($R_2$), BPG ($R_3$), TGE ($R_4$), and (s)-TGE ($R_5$) were selected from 13 candidates on the basis of their phase and solubility (Supplementary Fig. 1). Additionally, a liquid monomer, benzyl glycidyl ether, was selected for the comparison between solid- and solution-state mechanochemical polymerization. Further detail on the ball milling polymerization of liquid monomer is described in the following section. The successful synthesis of each monomer was confirmed via various NMR spectroscopy including $^1$H and $^{13}$C, correlation spectroscopy (COSY), and heteronuclear single quantum correlation (HSQC) (Supplementary Figs. 2–21), and their melting points were determined via differential scanning calorimetry (DSC) (Supplementary Figs. 22–26). It is of note that all monomers exhibited a proportional increase in melting point with increasing molecular weight, except for the BPG ($R_3$), which has a similar melting point but a lower molecular weight than TGE ($R_4$). Additionally, (s)-TGE ($R_5$) was used to compare the effect of melting point compared with TGE ($R_4$) at a fixed molecular weight. Furthermore, we investigated whether chirality of monomers affords control of the polymer tacticity via ball milling polymerization.

### Solid-state polymerization of polyethers via ball milling

After the successful characterization of various functional epoxide monomers, they were subjected to mechanochemical AROP in the presence of $t$-BuP$_4$ as the base and benzyl alcohol as the initiator using a vibratory ball mill (MM400) (See methods for details). Metal-free organic phosphazene superbase, $t$-BuP$_4$, was selected because it enables versatile access to a diverse array of polyethers with sensitive functional moieties at room temperature[21,22]. All mechanochemical polymerizations were targeted to a degree of polymerization (DP) of 50 using a stainless steel jar with stainless steel balls.

The chemical structures of all polymers were confirmed via $^1$H NMR spectroscopy (Fig. 2). The conversions of monomer-to-polymer for PMPG, PDPG, PBPG, PTGE, and (s)-PTGE were 47.7%, 64.1%, 86.5%, 90.8%, and 92.1%, respectively, as determined by the ratio of the methylene proton of each monomer to the polymeric protons at 6.87–6.61 (PMPG), 6.26–5.88 (PDPG), 4.20–3.71 (PBPG), 3.22–2.92 (PTGE), and 3.20–2.91 ppm ((s)-PTGE), respectively (a as shown in Supplementary Figs. 2, 6, 10, 14, and 18). A longer polymerization time of 2 h did not improve the conversion of (s)-TGE monomer (Supplementary Fig. 27). Further, the DP and corresponding number-average molecular weights ($M_{n,NMR}$) of resulting polyethers were calculated from the ratio of the protons of benzyl alcohol initiator (x) at 4.48 ppm to the polymeric protons at 4.21–3.45 (PMPG), 4.09–3.45 (PDPG), 3.22–2.92 (PTGE), 3.20–2.91 ppm ((s)-PTGE), respectively (Table 1, and Supplementary Notes. 1–4). Overall, unreliable $M_{n,NMR}$ values were obtained for PBPG because the benzyl initiator was not clearly visible, possibly due to screening of the biphenyl groups via strong π–π interactions. Additionally, the structures of the resulting polyethers were revealed by the $^{13}$C NMR spectra (Supplementary Figs. 28–32). Notably, the methine carbon of the (s)-PTGE polymer exhibited high isotacticity relative to that of the atactic PTGE polymer (Supplementary Figs. 31–33)[23]. This result clearly supports the absence of racemization during ball milling polymerization, which is often observed in solution polymerization.

The effects of various ball milling parameters, including the materials of the jar and balls, size of the balls, and number of balls, on mechanochemical AROP of the functional monomers were investigated (Supplementary Table 1). We observed that the jars and balls made of high-density materials induced higher monomer conversions attributed to higher mechanical forces. Moreover, larger sizes and numbers of balls afforded higher conversions due to greater impact per collision. We estimated that the filling degree of the milling balls in jar is approximately 27%, which is reported to be efficient for mixing[24].

The roles of the initiator and base in the polymerization system were also verified (Supplementary Figs. 34 and 35). Our initial attempt of mechanochemical polymerization in the absence of a superbase was

**Fig. 1 | Scheme for mechanochemical polymerization.** (Top) Mechanochemical anionic ring-opening polymerization (AROP) reaction of the functional epoxide monomers; (bottom) 4-methoxyphenyl glycidyl ether (MPG; $R_1$), 3,5-dimethoxyphenyl glycidyl ether (DPG; $R_2$), biphenyl glycidyl ether (BPG; $R_3$), trityl glycidyl ether (TGE; $R_4$), and (s)-trityl glycidyl ether ((s)-TGE; $R_5$) with corresponding physical properties of each monomer. mp: melting point, MW: molecular weight.

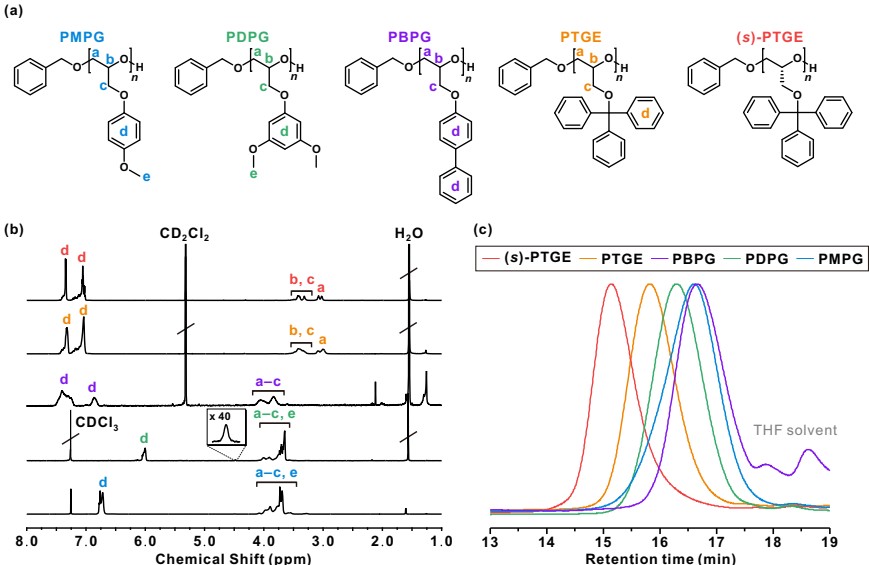

**Fig. 2 | Characterization of synthesized polyethers. a** chemical structures of polyethers synthesized via mechanochemical anionic ring-opening polymerization, **b** $^1$H NMR spectra of poly(4-methyoxyphenyl glycidyl ether) (PMPG, blue), poly(3,5-dimethoxyphenyl glycidyl ether) (PDPG, green), poly(biphenyl glycidyl ether) (PBPG, purple), poly(trityl glycidyl ether) (PTGE, orange), and poly((s)-trityl glycidyl ether) ((s)-PTGE, red), and **c** the GPC traces of PMPG (blue), PDPG (green), PBPG

(purple), PTGE (orange), and (s)-PTGE (red). $^1$H NMR spectra were obtained in CD$_2$Cl$_2$ and CDCl$_3$, and GPC was conducted in THF using a RI signal with polystyrene (PS) standards. Table 1 presents a detailed characterization of the synthesized polyethers. Note that the peak intensity of PBPG is considerably lower than the others due to the limited solubility in THF.

unsuccessful, whereas mechanochemical AROP of the (s)-TGE monomer in the absence of the benzyl alcohol initiator, proceeded owing to self-initiation, albeit with its relatively broader dispersity. These results indicate that bases play a crucial role in allowing the polymerization reaction to occur, while the initiator is important for the control of the mechanochemical AROP reaction.

The mechanistic details of mechanochemical AROP were revealed by analyzing the initiation steps via MALDI-ToF spectrometry (Supplementary Figs. 36–40). Each spectrum exhibited two sets of peaks separated by a regular interval, one of which corresponded to the molecular weight of the desired homopolymer chain end-capped with the benzyl alcohol initiator and the other corresponded to that of the homopolymer chain end-capped with the monomer alkoxide produced via elimination of the α-proton from the epoxide group of the monomer (Supplementary Fig. 41). This result is in good agreement with the one obtained in the absence of the benzyl alcohol initiator (Supplementary Fig. 35). In the case of PTGE and (s)-PTGE, additional series of peaks were observed due to decomposition of the trityl substituents during measurements (Supplementary Figs. 39 and 40). Specifically, a higher fraction of the peaks from the initiation by benzyl alcohol was observed in the polymerization of (s)-PTGE compared with self-initiation (Supplementary Fig. 42, and Supplementary Note 5). Considering the ambient conditions of the ball milling polymerization, interestingly, all the resulting polymers exhibited negligible initiation by water, which is one of the most challenging features of AROP.

The GPC traces of all polymers exhibited monomodal distributions with a narrow dispersity (Đ) of 1.12–1.21 (Fig. 2c), thereby demonstrating the controlled nature of the mechanochemical polymerization. This controlled nature is further demonstrated for (s)-TGE as a model monomer, which exhibited a distinct shift of the GPC trace toward a lower retention time with increasing reaction time and target DP (Supplementary Figs. 43 and 44, and Supplementary Tables 2 and 3). Overall, the $M_{n,GPC}$ values of all the resulting polyethers were lower than the $M_{n,th}$ due to high hydrophobicity of the epoxide monomers. It is worthy of noting that the isotactic (s)-PTGE polymer exhibited a higher $M_{n,GPC}$ of 5,950 g/mol compared with 3,720 g/mol for the atactic PTGE polymer with a similar $M_{n,th}$, in good agreement with

previous literature[25]. Meanwhile, the $M_{n,GPC}$ of PBPG is lower than that of the other polymers, possibly due to the strong π–π interactions that reduce the hydrodynamic volume.

Furthermore, to examine whether degradation of polyethers occurs during mechanochemical AROP, an (s)-PTGE polymer with a similar molecular weight to that prepared by mechanochemical polymerization was synthesized using the conventional solution polymerization, as confirmed via $^1$H NMR and GPC analyses (Supplementary Fig. 45). The prepared solid (s)-PTGE ($M_{n,GPC}$ = 4880 g/mol) was subjected to ball milling for an extended time beyond the mechanochemical AROP reaction. Considering almost identical GPC traces, we could exclude the possibility of backbone degradation during mechanochemical polymerization under the existing conditions (Supplementary Fig. 46).

DSC analysis was performed to determine the thermal properties of the synthesized polymers (Table 1 and Supplementary Fig. 47 and 48). The $T_g$ values generally increased with the number of phenyl substituents (i.e., PMPG < PDPG < PTGE). Moreover, PBPG exhibited exceptionally high crystallinity due to strong π–π interactions, which is consistent with the aforementioned NMR and GPC results. The observed difference in $T_g$ value between PTGE and (s)-PTGE polymers is also influenced by stereoregularity[26]. Although stereoregularity was controlled in the case of (s)-PTGE, $T_m$ value was not observed, possibly owing to the high steric hindrance of the trityl substituents (Supplementary Fig. 48).

## A comparison between solid-state and solution polymerizations

As reported recently, mechanochemical synthesis provides unexpected selectivity and reactivity compared with conventional approaches[8,27,28]. With this in mind, we were prompted to compare the reactivity of the selected functional epoxide monomers toward ball milling and conventional solution polymerization (Fig. 3). Figure 3a indicates that during solid-state mechanochemical polymerization, the conversion of monomer-to-polymer gradually increased with reaction time up to 30 min, demonstrating controlled mechanochemical polymerization. Interestingly, the bulkiest monomers, that is, TGE and (s)-

**Table 1 | Characterization of the synthesized polyethers via ball milling**

| Polymer | Conv.[a] (%) | $M_{n,th}$ (g/mol) | $M_{n,NMR}$ (g/mol) | $DP_{NMR}$ | $M_{n,GPC}$[d] (g/mol) | Đ[d] | $T_g$[e] (°C) | mp[f] (°C) | $T_{jar}$[h] (°C) |
|---|---|---|---|---|---|---|---|---|---|
| **PMPG** | 47.7 ± 4.8 | 6470 | 31790[b] | 176 | 2380 | 1.21 | 9.0 | 47.8 | 41.2 |
| **PDPG** | 64.1 ± 12.5 | 8050 | 8760[b] | 41 | 2760 | 1.17 | 13.1 | 63.8 | 44.5 |
| **PBPG** | 86.5 ± 0.2 | 9880 | n.d. | n.d. | 2100 | 1.12 | 204.4 [g] | 90.6 | 61.3 |
| **PTGE** | 90.8 ± 0.2 | 14080 | 23490[c] | 74 | 3720 | 1.13 | 74.0 | 87.8 | 56.8 |
| **(s)-PTGE** | 92.1 ± 1.0 | 14420 | 19180[c] | 60 | 5950 | 1.15 | 82.0 | 101.2 | 50.0 |

All polymerizations were targeted to a DP of 50 and conducted for 30 min.
[a]Monomer conversion as calculated from the 1H NMR spectrum of the crude monomer (*n* = 3, average values reported with standard deviation).
[b]Calculated from the 1H NMR spectrum of the isolated polymer (CDCl3, 400 MHz).
[c]Calculated from the 1H NMR spectrum of the isolated polymer (CD₂Cl₂, 400 MHz).
[d]Measured via GPC in THF using PS standard and RI signal.
[e]Determined via DSC at a rate of 10 °C/min.
[f]Melting point of the epoxide monomer measured via DSC.
[g]$T_m$.
[h]Temperature inside jar after ball milling reaction measured by IR thermometer.

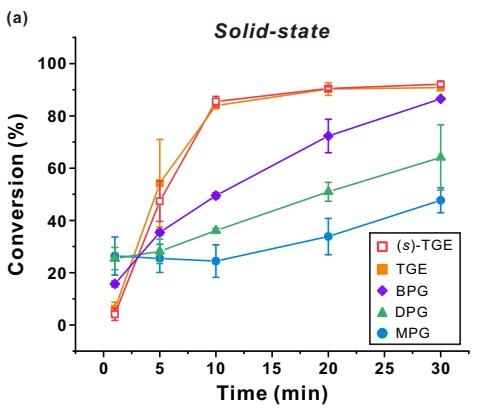

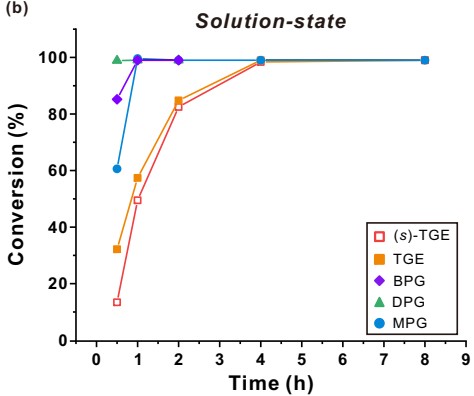

**Fig. 3 | Comparative monomer conversions of various functional epoxide monomers. a** Solid-state mechanochemical polymerization and (**b**) solution polymerization in toluene at 60 °C. Solid-state polymerization was performed in triplicates to minimize the batch-to-batch variation inherent in mechanochemical reactions.

TGE, exhibited the highest reactivities under mechanochemical polymerization, with the reactivities of the other less bulky monomers decreasing in the order of BPG > DPG > MPG. It is also noted that the conversion generally increased linearly with reaction time for the bulky monomers. The other monomers exhibited the following two reactivity regimes over time: (i) initial bulk polymerization via an unavoidable physical mixing process, which occurs during preparation of the milling jar with reactants and balls (particularly for MPG and DPG monomers with high reactivity in solution as evidenced in Supplementary Figs. 49–54) and (ii) mechanochemical polymerization that becomes dominant following the initial mixing process.

Despite the unavoidable initial bulk polymerization during ball milling, the polymerization kinetics of these monomers were further elucidated by conducting ¹H NMR measurements. Although the reaction conversion during the predominant bulk polymerization stage was excluded, the mechanochemical polymerization rate constants were determined by plotting $\ln([M]_0/[M]_t)$ against time (Supplementary Fig. 55). The linear correlation of these plots demonstrated the controlled mechanochemical AROP of five epoxide monomers with the following apparent propagation rate constant ($k_{p,app}$) for each monomer: $2.18 \times 10^{-2}$ min⁻¹ (MPG), $3.59 \times 10^{-2}$ min⁻¹ (DPG), $6.66 \times 10^{-2}$ min⁻¹ (BPG), $17.64 \times 10^{-2}$ min⁻¹ (TGE), and $17.91 \times 10^{-2}$ min⁻¹ ((s)-TGE). These values are considerably higher than those previously observed for conventional AROP of dimethyl oxazoline glycidyl ether ($1.83 \times 10^{-2}$ min⁻¹)[29] and azidoethyl glycidyl ether ($1.24 \times 10^{-2}$ min⁻¹)[30], indicating a significant acceleration in the mechanochemical polymerization reaction.

Independently, conventional solution-phase polymerization was performed for the same monomers as a control. Considering the heat generated during the ball milling process, the corresponding solution reactions were performed in toluene at 60 °C (Supplementary Fig. 56). In this case, as expected, the monomers with less steric hindrance in the side chain exhibited a faster conversion rate (Fig. 3b), which is consistent with a previous study[22]. Moreover, conventional AROP exhibited a considerably longer reaction time than ball milling, and, most interestingly, the trend in monomer reactivity was clearly reversed. In addition, when the monomers were polymerized in bulk free of solvent at 60 °C, the reactivity trend was identical to that of solution polymerization (Supplementary Table 4). Conversely, when the liquid monomer, that is, benzyl glycidyl ether, with a moderate steric hindrance was subjected to ball milling and solution polymerizations, no difference in reactivity was observed, suggesting that the unique reactivity is only displayed with solid monomers (Supplementary Figs. 57 and 58). Taken together, these findings demonstrate that solid-state mechanochemical AROP exhibits unique reactivity compared with conventional polymerization techniques.

Additionally, we tried to control the unavoidable heat generation by using temperature-controllable ball-milling equipment (Supplementary Fig. 59)[31]. It was found that the reactivity does not change considerably even when the temperature is under control compared to ball milling reaction using MM 400 (Supplementary Table 4, and Supplementary Fig. 60). This observation proves that the ball milling reactivity is not significantly affected by heat.

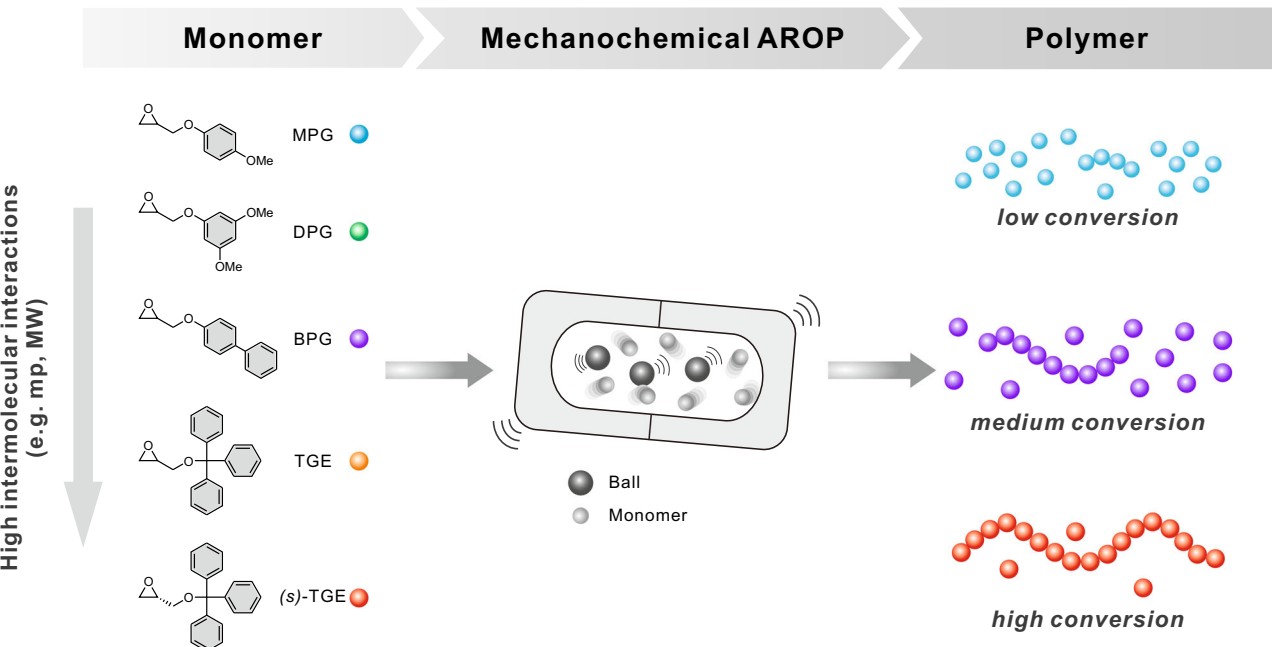

**Fig. 4 | Schematic summary of this study.** An illustration of the mechanochemical polymerization reactivity of functional epoxide monomers with respect to their molecular structures and physical properties.

## Exploration of the influential parameters of mechanochemical AROP reactivity

Inspired by the unique reactivity during mechanochemical polymerization, we were prompted to correlate the reactivity with the physical properties of the monomers. Figure 4 depicts the effect of monomer structure on its mechanochemical reactivity. Moreover, as the intermolecular interactions of the monomers could be deduced from their melting points and molecular weights, the corresponding reactivity trends during ball milling polymerization could be explored by plotting the conversion against melting point and against molecular weight at various reaction times (Supplementary Fig. 61). Surprisingly, as mechanochemical polymerization progresses, close linear correlations were found between the conversion and both the melting points and molecular weights of the monomers. Although it is challenging to decouple the physical properties of monomers individually, this result suggests that the melting point of monomers, which integrates various intermolecular interactions, is the most critical predictor of ball milling polymerization reactivity. It can be proposed that the reactivity reversal observed in the solid-state polymerization is possibly originated from that the bulky substituents with a higher intermolecular interaction in the monomer (e.g. melting point) can recruit more mechanochemical forces during the ball-milling polymerization. On the other hand, the bulky substituent could cause retardation of the polymerization in solution due to steric hindrance. However, it should be considered that quantitative analysis of the reactivity is only valid when the reaction temperature is constant during mechanochemical reactions of different monomers Nonetheless, we believe that proposing a semiempirical relationship in deciphering and/or predicting the reactivity of the monomers during the mechanochemical polymerization by itself deserves its own merit, as the field of mechanochemical polymerization is still in its infancy and rapidly growing field.

## Discussion

Herein, the ball milling process was applied to mechanochemical AROP of five functional epoxide monomers with distinct physical properties. The successful syntheses of the desired polyethers with controlled molecular weight and dispersity were characterized via a series of $^1$H, $^{13}$C NMR, GPC, DSC, and MALDI-ToF analyses. The effects of various ball milling parameters demonstrated that mechanochemical force plays a critical role in the AROP. Moreover, the mechanochemical polymerization proceeded without degradation of the resulting polyether during the ball milling reaction.

We found the unique mechanochemical reactivity of functional epoxide monomers in mechanochemical AROP, consistent with recent studies demonstrating the unique chemical reactivity associated with mechanochemical reactions. Specifically, the bulkiest monomer exhibited a faster conversion during mechanochemical polymerization, whereas monomers with less steric hindrance exhibited lower reactivity. However, a liquid epoxide monomer with moderate steric hindrance revealed that the unique ball milling reactivity only applies to solid-state monomers. Moreover, the propagation rate constant for ball milling AROP was considerably higher than those obtained for the conventional AROP process, indicating significant acceleration for ball milling polymerization of solid-state monomers.

Finally, to obtain further insight into the unique reactivity, the influential parameters for ball milling polymerization reactivity were investigated based on the physical properties of the monomers, including the melting point and molecular weight. Most notably, a linear correlation between the ball milling reactivity and melting point of the monomers was observed, implying that the melting point of monomers is a critical predictor of ball milling polymerization reactivity. We anticipate that this study will contribute to understanding the fundamental principles and future applications of mechanochemical polymerization.

## Methods
### Materials

Glycidol, triethylamine (NEt$_3$), tetrabutylammonium hydrogensulfate (TBAHSO$_4$), phosphazene base $t$-BuP$_4$ solution (0.8 M in hexane), anhydrous toluene, and anhydrous benzyl alcohol were purchased from Sigma-Aldrich. Epichlorohydrin (ECH), 3,5-dimethoxyphenol trityl chloride, 4-phenyl phenol, ($s$)-trityl glycidyl ether (($s$)-TGE), and 4-methoxyphenyl glycidyl ether (MPG) were obtained from Tokyo Chemical Industry. Moreover, 4-dimethylaminopyridine (DMAP) was purchased from Alfa Aesar. Potassium hydroxide (KOH) flakes and

ethyl acetate (EtOAc) were obtained from Daejung. Anhydrous magnesium sulfate ($MgSO_4$) was purchased from Duksan. Furthermore, (s)-TGE and MPG were purified via azeotropic distillation using toluene cosolvent to remove moisture prior to polymerization. All other reagents were used as received without further purification. The deuterated NMR solvents $CDCl_3$ and $CD_2Cl_2$ were purchased from Cambridge Isotope Laboratory.

## Characterization

The $^1H$ and $^{13}C$ NMR spectra were recorded at 25 °C on Bruker 400- and 800-MHz (at Korea Basic Science Institute) spectrometers, using $CDCl_3$ ($\delta H = 7.26$ ppm and $\delta C = 77.16$ ppm), $CD_2Cl_2$ ($\delta H = 5.32$ ppm and $\delta C = 54.00$ ppm), and acetone ($\delta C = 206.26$ ppm, and 29.84 ppm) as internal standards. The solid-state $^{13}C$ magic-angle spinning NMR spectrum was recorded on a Bruker WB PLUS AVANCE II + 400-MHz NMR spectrometer at the Korea Basic Science Institute Seoul Western Center. Differential scanning calorimetry (DSC) was performed using a Q200 model (TA instruments) between −80 °C and 150 °C at a heating and cooling rate of 10 °C min$^{-1}$ under a nitrogen ($N_2$) atmosphere. After annealing, a second cycle was used to determine the melting points of the epoxide monomers and the thermal properties of the polymers. Gel permeation chromatography (GPC) was performed using an Agilent 1200 Series equipped with an autoinjector and refractive index (RI) detector using tetrahydrofuran (THF) as the eluent at 25 °C and a flow rate of 1.00 mL min$^{-1}$. The number- and weight-averaged molecular weights ($M_n$ and $M_w$) and molecular weight distributions ($M_w/M_n$, Đ) were calculated using polystyrene (PS) standards with peak molecular weights ($M_p$) of 250–1,100,000 g mol$^{-1}$ provided by Sigma-Aldrich. MALDI-ToF mass spectrometry was performed using a Bruker Autoflex Max mass spectrometer. Ball milling experiments were performed using a Retsch Mixer Mill MM400 laboratory mill. All monomers were subjected to azeotropic distillation with anhydrous toluene prior to polymerization to ensure high purity.

## Synthesis of 3,5-dimethoxyphenyl glycidyl ether (DPG)

In a 500-mL round-bottomed flask, a 40% aqueous KOH solution was prepared by dissolving 18.18 g of KOH in 27.27 mL of water at 0 °C. Subsequently, TBAHSO$_4$ (0.55 g, 1.62 mmol) and ECH (12.70 mL, 162.00 mmol) were added to the KOH solution and stirred for 30 min. Thereafter, 3,5-dimethoxyphenol (5.00 g, 32.40 mmol) in toluene (31.53 mL, toluene:water = 1:1 v/v) was added dropwise. The reaction mixture was stirred for 18 h at 25 °C, and the reaction progress was monitored via thin-layer chromatography (TLC). After diluting with water, the mixture was extracted using EtOAc. The combined organic layers were washed with brine, dried using anhydrous $MgSO_4$, and concentrated in vacuo. The crude product was purified via flash column chromatography using acetone/hexane (1:8 v/v) as the eluent. Further purification was performed via azeotropic distillation of the column product using toluene cosolvent to remove any moisture, and the pure monomer was obtained as a white solid (3.47 g, 51%): m.p. 64 °C; $^1H$ NMR (400 MHz, $CDCl_3$) δ 6.10 (s, 3H), 4.19 (dd, $J = 11.0$, 3.1 Hz, 1H), 3.91 (dd, $J = 11.0$, 5.7 Hz, 1H), 3.77 (s, 6H), 3.39–3.29 (m, 1H), 2.91 (dd, $J = 4.8$, 4.2 Hz, 1H), 2.75 (dd, $J = 4.9$, 2.7 Hz, 1H) (Supplementary Fig. 6); $^{13}C$ NMR (101 MHz, $CDCl_3$) δ 161.59 (s), 160.41 (s), 93.55 (s), 93.49 (s), 68.85 (s), 55.46 (s), 50.15 (s), 44.84 (s) (Supplementary Fig. 7).

## Synthesis of biphenyl glycidyl ether (BPG)

TBAHSO$_4$ (0.50 g, 1.47 mmol) and ECH (11.53 mL, 147.00 mmol) were added to the KOH solution and stirred for 30 min. Subsequently, 4-phenylphenol (5.00 g, 29.40 mmol) was added dropwise. The reaction mixture was stirred for 18 h at 25 °C, and the reaction progress was monitored via TLC. After diluting with water, the mixture was extracted using EtOAc. The combined organic layers were washed with brine, dried using anhydrous $MgSO_4$, and concentrated in vacuo. The crude product was purified via flash column chromatography using acetone/

hexane (1:10 v/v) as the eluent. Further purification was performed via azeotropic distillation of the column product with toluene cosolvent to remove any moisture, and the pure monomer was obtained as a white solid (3.50 g, 52%): m.p. 91 °C; $^1H$ NMR (400 MHz, $CDCl_3$) δ 7.57 (dd, $J = 12.9$, 5.1 Hz, 4H), 7.44 (t, $J = 7.6$ Hz, 2H), 7.32 (dd, $J = 14.4$, 7.0 Hz, 1H), 7.02 (d, $J = 8.8$ Hz, 2H), 4.30 (dd, $J = 11.0$, 3.2 Hz, 1H), 4.03 (dd, $J = 11.0$, 5.6 Hz, 1H), 3.42 (dd, $J = 5.6$, 4.2 Hz, 1H), 3.00–2.91 (m, 1H), 2.81 (dd, $J = 4.9$, 2.7 Hz, 1H) (Supplementary Fig. 10); $^{13}C$ NMR (101 MHz, $CDCl_3$) δ 158.03 (s), 140.68 (s), 134.34 (s), 128.75 (s), 128.22 (s), 126.77 (s), 114.92 (s), 68.84 (s), 50.19 (s), 44.78 (s) (Supplementary Fig. 11).

## Synthesis of trityl glycidyl ether (TGE)

Trityl glycidyl ether (TGE) was synthesized using a procedure reported by Güclü et al.[32] For subsequent polymerization, TGE was purified via azeotropic distillation using toluene cosolvent to remove any moisture, and the pure monomer was obtained as a white solid (2.42 g, 51%): m.p. 88 °C; $^1H$ NMR (400 MHz, $CD_2Cl_2$) δ 7.45 (dt, $J = 8.5$, 1.9 Hz, 6H), 7.36–7.29 (m, 6H), 7.29–7.22 (m, 3H), 3.33 (dd, $J = 10.7$, 2.8 Hz, 1H), 3.12 (ddt, $J = 5.5$, 4.1, 2.7 Hz, 1H), 3.03 (dd, $J = 10.7$, 5.7 Hz, 1H), 2.74 (dd, $J = 5.1$, 4.2 Hz, 1H), 2.56 (dd, $J = 5.1$, 2.7 Hz, 1H) (Supplementary Fig. 14); $^{13}C$ NMR (101 MHz, $CDCl_3$) δ 143.66 (s), 128.76 (s), 128.01 (s), 127.06 (s), 86.69 (s), 64.86 (s), 50.98 (s), 44.75 (s) (Supplementary Fig. 15).

## General procedure for synthesis of functionalized polyethers in the solid state

The syntheses of all polyethers in the solid state were performed using the ball milling technique. All mechanochemical polymerizations were targeted to a degree of polymerization (DP) of 50 using a 10-mL stainless steel jar with three stainless steel balls with a diameter of 12 mm. All ball milling experiments were performed in triplicate. Taking PDPG as an example (entry 2 in Table 1), benzyl alcohol (3.95 μL, 1.0 equiv) and phosphazene base $t$-BuP$_4$ solution (47.6 μL, 1.0 equiv., 0.8 M in hexane) were added to a 10-mL stainless steel screw cap jar with three 12-mm stainless steel balls under an Ar atmosphere. Subsequently, the purified DPG monomer (400.0 mg, 50.0 equiv) was added, and the stainless-steel jar was sealed with Teflon tape under an Ar atmosphere and placed in a vibratory ball-mill machine (MM400). The grinding reaction was allowed to proceed for 30 min at a frequency of 30 Hz. The reaction was then terminated by opening the jar in air and adding excess methanol. The product in the jar was then dissolved in dichloromethane and concentrated in vacuo. A portion of the mixture was used for conducting $^1H$ NMR spectroscopy to determine the conversion. The residual mixture was treated with Amberlite IR-120(H) to remove phosphazene base $t$-BuP$_4$. The product was then dissolved in dichloromethane and further purified via precipitation in cold methanol. The solution was then centrifuged at 5,000 rpm for 10 min to obtain the PDPG polymer as a transparent oil of the PDPG polymer. After purification, chemical structure, molecular weight, and Đ were determined via $^1H$ NMR and GPC measurements. $^1H$ NMR (400 MHz, $CDCl_3$) δ 6.03 (d, 123H), 4.48 (s, 2H), 4.09–3.45 (m, 451H) (Fig. 2); $^{13}C$ NMR (101 MHz, acetone-$d_6$) δ 162.49 (s), 161.70 (s), 94.11 (s), 93.77 (s), 79.04 (s), 70.86–69.97 (m), 68.89 (s), 55.50 (s) (Supplementary Fig. 29).

## Poly(4-methoxyphenyl glycidyl ether) (PMPG)

Transparent oil: $^1H$ NMR (400 MHz, $CDCl_3$) δ 6.87–6.61 (m, 451H), 4.49 (s, 2H), 4.21–3.45 (m, 896H) (Fig. 2); $^{13}C$ NMR (101 MHz, acetone) δ 154.81 (s), 153.90 (s), 116.34 (s), 115.33 (s), 79.20 (s), 70.69 (s), 69.52 (s), 55.77 (s) (Supplementary Fig. 28).

## Poly(biphenyl glycidyl ether) (PBPG)

Because of low solubility, the PBPG polymer was purified by washing it three times each with hexane, DCM, acetone, and methanol to remove the phosphazene base $t$-BuP$_4$ and any residual monomer. After purification, the product was obtained as a white solid: $^1H$ NMR (400 MHz,

$CDCl_3$) δ 7.40 (s, 7H), 6.86 (s, 2H), 3.94 (d, J = 89.5 Hz, 5H) (Fig. 2); Solid-state $^{13}C$ MAS NMR (101 MHz) δ 157.80 (s), 144.99–104.46 (m), 76.87 (s) (Supplementary Fig. 30).

### Poly(trityl glycidyl ether) (PTGE)

White solid: $^1H$ NMR (400 MHz, $CD_2Cl_2$) δ 7.68–6.86 (m, 1,065H), 4.33 (s, 2H), 3.42 (m, 213H), 3.04 (d, J = 14.4 Hz, 142H) (Fig. 2); $^{13}C$ NMR (201 MHz, $CD_2Cl_2$) δ 144.72 (s), 129.17 (s), 128.30 (s), 127.42 (s), 86.93 (s), 80.22–78.99 (m), 71.30–70.04 (m), 64.52–63.52 (m) (Supplementary Fig. 31).

### (s)-Poly(trityl glycidyl ether) ((s)-PTGE)

White solid: $^1H$ NMR (400 MHz, $CD_2Cl_2$) δ 7.49–6.92 (m, 810H), 4.30 (s, 2H), 3.36 (d, J = 38.0 Hz, 162H), 3.04 (d, J = 13.9 Hz, 108H) (Supplementary Fig. 27); $^{13}C$ NMR (201 MHz, $CD_2Cl_2$) δ 144.66 (s), 129.17 (s), 128.30 (s), 127.44 (s), 86.93 (s), 79.74 (s), 70.61 (s), 64.17 (s) (Supplementary Fig. 32).

### General procedure for the synthesis of functionalized polyethers in the solution state

All polyethers in the solution state were synthesized using the Schlenk technique under an Ar atmosphere with flame-dried glass tubes. For example, in the preparation of DPG, benzyl alcohol (3.95 μL, 1.0 equiv) and phosphazene base $t$-$BuP_4$ solution (47.6 μL, 1.0 equiv, 0.8 M in hexane) were dissolved in toluene (0.76 mL). The purified solid DPG monomer (400.0 mg, 50.0 equiv) was then added to the solution. The reaction was monitored via $^1H$ NMR spectroscopy by transferring a portion of the reaction mixture to determine the completion of the reaction. After confirming the disappearance of the residual epoxide signals of the monomer, the reaction was terminated by adding excess methanol. The mixture was then concentrated in vacuo and treated with Amberlite IR-120(H) to remove the phosphazene base $t$-$BuP_4$. The product was then dissolved in DCM and further purified via precipitation in cold methanol. The solution was then centrifuged at 5000 rpm (× 2800 g) for 10 min to obtain the PDPG polymer as a transparent oil. After purification, the chemical structure, molecular weight, and Đ were determined via $^1H$ NMR and GPC measurements.

### Degradation test of polyethers in the ball milling system

For this test, the (s)-PTGE polymer (400.0 mg) was added to a 10-mL stainless steel screw cap jar with three 12-mm stainless steel balls under ambient conditions. The jar was then locked and placed in the vibratory ball-mill machine, and the grinding reaction was allowed to proceed for 30 min at a frequency of 30 Hz. Aliquots were collected at intervals of 1, 5, 10, 20, and 30 min and analyzed via GPC measurement to determine $M_{n,GPC}$.

## Data availability

All the data analyzed in this research are included in this manuscript and Supplementary Information. All relevant data are available from the corresponding author upon request.

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

## Acknowledgements
This work was supported by the Samsung Research Funding & Incubation Center of Samsung Electronics under Project Number SRFC-MA1902-05 (B.-S.K) and by the National Research Foundation of Korea (NRF-2021R1A2C3004978 and NRF-2018R1A5A1025208) (B.-S.K).

## Author contributions
B.S.K. designed and supervised the research. J.H.P, and B.S.K prepared the draft of the paper. J.H.P, and K.A.H performed the experiments and analyses.

## Competing interests
The authors declare no competing interests.
