## [Peer Review File · Nature Communications]

Anionic Ring-Opening Polymerization of Functional Epoxide Monomers in the Solid StateReviewers' Comments:

Reviewer #1:

Remarks to the Author:

In this study, the author examines the correlation between monomer reactivity and their melting points in the mechanochemical anionic polymerization of epoxides and finds a trend in reactivity that is opposite to the usual expectation. It is already known that in mechanochemical reactions using ball milling, the reactivity of solid substrates tends to be less reactive than liquid substrates and roughly correlates with their melting points. In this study, epoxide monomers with different melting points are synthesized, and anionic polymerization is performed with a strong organic base (to investigate the differences in reactivity). Interestingly, contrary to the usual trend, the monomers with higher melting points showed higher reactivity (Figure 3a). In a control experiment using a solution system, the reactivity of monomers with higher melting points was relatively low; almost the opposite result was obtained.

While this is a very interesting result, there is one very significant concern. Reactivity is strongly influenced by the reaction temperature, but not all experiments were performed at a constant temperature (Figure S58). The differences in reaction rates between the monomers obtained are not large enough to make the temperature difference (40-60 degrees C) negligible. Solid monomers tend to have higher temperatures in the ball mill (Figure S58). Solid monomers are more strongly affected by friction and may generate more frictional heat than liquid monomers. If the reaction was accelerated by frictional heat, this is not a very interesting result. To clarify the results of this study, it is necessary to separate this effect of frictional heat from that of reaction heat. Experiments should be performed using a ball mill that can maintain a constant temperature in the jar, for example, the Retsch MM 500 control. Unless there are new experimental results on this temperature effect, this paper is not acceptable.

Another concern is that the author does not discuss at all why solid monomers are more reactive than liquid monomers. An explanation or hypothesis at the molecular level should be offered. For example, in topochemical reactions, the spatial proximity of reactive centers can explain reactivity. Single crystal X-ray structure analysis of monomers may provide a clue.

The author uses the term "rigidity" a lot as a property of a reactive solid monomer, but the correlation obtained is between melting point and reactivity. Rigidity is related to the mechanical strength of the solid monomer, which is a different concept from the melting point. The author states, "Moreover, as the rigidity of the monomers could be deduced from their melting points" (p13, line 257), which I think is not obvious. Either the use of the term "rigidity" needs to be revisited, or the mechanical strength of the monomer needs to be measured in a different way and correlated with reactivity or melting point.

Reviewer #2:

Remarks to the Author:

As the authors describe it in their introduction, mechanochemical polymerization reactions have grown considerably in the last year, but a basic understanding of the process itself is still missing. Thus, the approach to correlate melting point to reactivity the authors apply is interesting. The authors achieve this by the careful selection of different monomers and the characterization of the corresponding polymers by multiple analysis methods. Additionally, they complement this by a study of the same polymerization in solution, where they observe the classical, thus reversed, influence of rigidity. On a smaller side note, they also confirm that stereoregularity is retained during the synthesis and no racemization is taking place.

The methods applied by the authors are scientifically sound and the conclusions are supported by sufficient data. Several experiments were repeated in triplicates and thus the data should be reliable.

The figures support the conclusions, and the text is comprehensible and well-structured. The paper is of significant interest to the polymer community as a whole and the mechanochemical community in particular.

Major Points:

P1: The NMR of (s)-PTGE and the GPC trace of PBPG have been omitted in Figure 2. Why have they been left out?

P2: t-BuP4 was used in this study. In recent publications in mechanochemical polymerizations DBU and TBD, two more commonly found bases, have been found to achieve good results. The authors should elaborate on why only one base was considered in this manuscript.

P3: The authors claim that increasing the number of milling balls in the vessel is increasing the impact per collision. This is not correct. Due to the hindering of each other, more milling balls, in fact, lead to a reduced free path length of the milling balls and thus lower overall speed and smaller impact energy. The higher yield, however, is an interesting phenomenon, that in my opinion is rather due to the increased number of collisions and thus a higher number of shear events leading to more reactive surfaces being created and better mixing. There has been a lot of work done by the mechanochemical community and the early works of Ondruschka and more recent works of Qwade and Boldyreva deal with this topic.

Minor Points:

pI: Figure S27 assignment on the structure is missing (a,b,c, etc.)

pII The authors talk about the selection of 5 monomers and explain their reasoning well. At first, I was disappointed because no liquid monomer was used, however, in the end (line 238 ff.) they discuss the reaction with a liquid monomer. I think they should mention in their selection process, that a control experiment with a liquid monomer has been made.

Suggestions for further work:

In line 213ff the authors discuss the fact, that they cannot control the first reaction regime, because an initial mixing during the preparation of the milling jar is unavoidable. Several mechanochemists have had the same problem and in some of their works, Hernandez and Bolm used glass capillaries to seal one of their reactants which only breaks when the milling is initialized. Especially for the field of controlled polymerization of soluble polymers, this technique could be highly beneficial.

Reviewer #3:

Remarks to the Author:

The authors present an anionic ring-opening polymerization of functional epoxide monomers in the solid state using a mechanochemistry approach. The authors synthesized of polyethers from diverse functional epoxide monomers using solid-state and solution polymerization reactions. To understand the distinctive reactivity of ball milling, the key characteristics of the monomers were examined. The reaction conditions were optimized with different types of jars and balls as well as different times. The results showed bulky monomers exhibited faster conversions in the solid-state contrast to that observed for solution polymerization. This was confirmed by NMR, GPC, and MALDI-ToF analyses. The authors showed linear correlation was observed between the conversion of the ball milling polymerization and the melting point of the functional epoxide monomers, indicating the melting point is a significant factor of mechanochemical polymerization reactivity. Overall, this is a well-written and organized manuscript highlighting a better understanding of ring-opening polymerization synthesis using the mechanochemistry approach. So, this manuscript is worthy of publication in the Nature Communication Journal with revisions.

The authors should address the comments below:

1- In Table 1 and Supplementary Table 1, I would like to see DP from the ¹HNMR spectrum for each isolated polymer. In addition, it's worth adding the Initiator efficiency (IE%) of the BnOH from the crude ¹HNMR spectrum for each polymer which has been obtained by different researchers such as Nelson and coworkers, *Macromolecules* 2022, 55, 21, 9740.

2- On page 6, line 102, Table 1 should be placed directly after the end of this paragraph.

- 3- On page 7, there is no data provided for the control reactions such as M_n , DP, Conv% and IE%, so, for a clear comparison with results from Table 1, I would like to see a table for these reactions.
- 4- In Table 1, the data shows that $M_{n,nmr}$ values for all polymers were much higher than $M_{n,GPC}$, So, I am wondering about the reason for this.
- 5- The authors discuss conversion %, $M_{n,th}$, and $M_{n,NMR}$, but never define it or explain how they calculated it. Please define these parameters in addition to the DPNMR as well as IE% by using the equation and explaining how they calculated it.
- 6- In the introduction, by the end of line 30, a reference for this paragraph should be added.
- 7- In Supplementary Information, a Table of content needs to be added.

Reviewer(s)' Comments to Author:

Reviewer: 1

Comments:

In this study, the author examines the correlation between monomer reactivity and their melting points in the mechanochemical anionic polymerization of epoxides and finds a trend in reactivity that is opposite to the usual expectation. It is already known that in mechanochemical reactions using ball milling, the reactivity of solid substrates tends to be less reactive than liquid substrates and roughly correlates with their melting points. In this study, epoxide monomers with different melting points are synthesized, and anionic polymerization is performed with a strong organic base (to investigate the differences in reactivity). Interestingly, contrary to the usual trend, the monomers with higher melting points showed higher reactivity (Figure 3a). In a control experiment using a solution system, the reactivity of monomers with higher melting points was relatively low; almost the opposite result was obtained.

1) While this is a very interesting result, there is one very significant concern. Reactivity is strongly influenced by the reaction temperature, but not all experiments were performed at a constant temperature (Figure S58). The differences in reaction rates between the monomers obtained are not large enough to make the temperature difference (40-60 degrees C) negligible. Solid monomers tend to have higher temperatures in the ball mill (Figure S58). Solid monomers are more strongly affected by friction and may generate more frictional heat than liquid monomers. If the reaction was accelerated by frictional heat, this is not a very interesting result. To clarify the results of this study, it is necessary to separate this effect of frictional heat from that of reaction heat. Experiments should be performed using a ball mill that can maintain a constant temperature in the jar, for example, the Retsch MM 500 control. Unless there are new experimental results on this temperature effect, this paper is not accept.

We appreciate reviewer for this critical comment. It should be first emphasized that here we are comparing the reactivity of solid monomers during anionic ring-opening polymerization. It is true indeed that the decoupling the effect of frictional heat from reaction heat is critical to clarify the results of this study. While we do not have the Retsch MM 500 control at the moment, we modified the ball milling instrument (i.e., Retsch MM 400) using a cooling jacket for the reaction vessel that is connected to a chiller to maintain the constant reaction temperature (see the recent report of using the cooling jacket in Min et al. "Mechanochemical Direct Fluorination of Unactivated C(*sp*³)-H Bonds" *Adv. Synth. Catal.* **2022**, 364, 1975).

Even though the temperature still rose after the ball milling reactions, the degree of temperature change was considerably lower value of about 17–28 °C than those observed in the previous

setup without temperature control. It was confirmed that the reactivity order of solid epoxide monomers does not change, which is consistent with previous result. This observation proves that the ball milling reactivity is not significantly affected by frictional heat generated by the solid monomers. The results of this study along with the experimental setup with a cooling jacket is included in the revised manuscript and supplementary information.

[Page 13 in the revised manuscript] – new sentence included is highlighted

Taken together, these findings demonstrate that solid-state mechanochemical AROP exhibits unique reactivity compared with conventional polymerization techniques.

Additionally, we performed the mechanochemical polymerization using a modified ball milling instrument using a cooling jacket for the reaction vessel to maintain the constant reaction temperature (Supplementary Fig. 59 and 60, and Supplementary Table 4).³¹ In concert with the previous results, the reactivity order of solid epoxide monomers does not change, suggesting the frictional heat generated by the solid monomers does not influence the ball milling reactivity significantly.

[Page 14 in the revised manuscript] – new reference is added

31. Min, S., Park, B., Nedsaengtip, J., Hong, S-H. Mechanochemical direct fluorination of unactivated C(*sp*³)-H bonds. *Adv. Synth. Catal.* **364**, 1975–1981 (2022).

[Page S-63 in the revised Supplementary Information] – revised Supplementary Table 4

Supplementary Table 4. Comparison of monomer conversions under different experimental setup: ball milling, solution, and bulk polymerization.

Polymer	Conv. ^a (%)			
	Ball milling		Solution ^b	Bulk ^c
	ambient	w/ cooling jacket		
PMPG	47.7 ± 4.8	44.8	60.6	98.3
PDPG	64.1 ± 12.5	55.0	98.9	87.7
PBPG	86.5 ± 0.2	78.7	85.2	10.1
PTGE	90.8 ± 0.2	91.7	32.2	0
(s)-PTGE	92.1 ± 1.0	91.9	13.4	0

All polymerizations were targeted to a DP of 50 and conducted for 30 min. ^aMonomer conversion as calculated from the ¹H NMR spectrum of the crude monomer. ^bSolution polymerization in 2.5 M toluene at 60 °C. ^cBulk polymerization at 60 °C.

[Page S-66 and S-67 in the revised Supplementary Information] – new Supplementary Fig. S59 and S60 are included

Supplementary Fig. 59. Temperature-controllable ball-milling equipment used in this study; cooling jackets (left), MM400 with cooling jackets (center), and overall mechanochemical polymerization setup with chiller (right).

Supplementary Fig. 60. IR thermometer images showing temperature inside the jar after polymerization using (a) Ball-milling using MM400 under ambient condition and (b) temperature-controllable ball-milling MM400. All reactions were performed for 30 min at 30 Hz. The average temperature difference between two setup was determined to be around 17–28 °C

2) Another concern is that the author does not discuss at all why solid monomers are more reactive than liquid monomers. An explanation or hypothesis at the molecular level should be

offered. For example, in topochemical reactions, the spatial proximity of reactive centers can explain reactivity. Single crystal X-ray structure analysis of monomers may provide a clue.

We thank the reviewer for this critical comment. We feel, however, that there is a misunderstanding of the reactivity order. In fact, our study indicates that the liquid monomer is more reactive than the solid monomer – as commented in the 4th line of the reviewer's report. Because of the homogeneous nature of the liquid monomers during the polymerization, the liquid monomers displayed a higher reactivity over the solid monomers in general and even in mechanochemical polymerization.

Furthermore, the mechanochemical reaction of liquid monomer is homogeneous in its nature, showing similar reactivity to the conventional method. On the other hand, solid monomer experiences the heterogeneous reaction, leading to their unique reactivity in mechanochemistry which could not be observed in bulk polymerization (see Supplementary Table 4). It is obviously known that the homogeneous system result in faster reactivity than heterogeneous system.

Meanwhile, as suggested by the reviewer, the topochemical reaction is a chemical transformation generated by alignment of reactive centers in molecules under the crystalline state. However, all solid monomers prepared in this study do not possess any sign of crystallinity as determined in the DSC thermograms (see Supplementary Fig. 22–26). This result provides that these solid monomers are amorphous in nature, thus limiting to access the single crystal X-ray analysis.

3) The author uses the term "rigidity" a lot as a property of a reactive solid monomer, but the correlation obtained is between melting point and reactivity. Rigidity is related to the mechanical strength of the solid monomer, which is a different concept from the melting point. The author states, "Moreover, as the rigidity of the monomers could be deduced from their melting points" (p13, line 257), which I think is not obvious. Either the use of the term "rigidity" needs to be revisited, or the mechanical strength of the monomer needs to be measured in a different way and correlated with reactivity or melting point.

We appreciate reviewer for comment. In this study, we used "rigidity" as a collective term to represent high physical and chemical properties of monomers, including melting point, molecular weight, and possibly all other related inter- and intramolecular interactions. It does not necessarily indicate that the monomers with high mechanical strength. Other terms such as resilient, robust, hard, and solid were also considered; however, we found that the "rigidity" was the most appropriate term to correlate the mechanochemical reactivity of the solid monomers under ball milling polymerization. While two other reviewers did not raise the concern on the term "rigidity", we are certainly open to the suggestion of other term to replace the "rigidity".

Reviewer: 2

Comments:

As the authors describe it in their introduction, mechanochemical polymerization reactions have grown considerably in the last year, but a basic understanding of the process itself is still missing. Thus, the approach to correlate melting point to reactivity the authors apply is interesting. The authors achieve this by the careful selection of different monomers and the characterization of the corresponding polymers by multiple analysis methods. Additionally, they complement this by a study of the same polymerization in solution, where they observe the classical, thus reversed, influence of rigidity. On a smaller side note, they also confirm that stereoregularity is retained during the synthesis and no racemization is taking place.

The methods applied by the authors are scientifically sound and the conclusions are supported by sufficient data. Several experiments were repeated in triplicates and thus the data should be reliable. The figures support the conclusions, and the text is comprehensible and well-structured. **The paper is of significant interest to the polymer community as a whole and the mechanochemical community in particular.**

1) The NMR of (*s*)-PTGE and the GPC trace of PBPG have been omitted in Figure 2. Why have they been left out?

We appreciate the reviewer for the careful comment. As the ¹H NMR spectrum of (*s*)-PTGE was almost identical to PTGE and the GPC trace of PBPG was very low in signal due to the low solubility, we placed them in the Supplementary Information (Figure S27 and Figure S44 in the original manuscript). As suggested, we included the aforementioned data in Figure 2 for the comparison with other polymers in the revised manuscript.

[Page 7 in the revised manuscript] – Revised Figure 2 is included and the new sentence is highlighted

Figure 2. Characterization of synthesized polyethers. (a) chemical structures of polyethers synthesized via mechanochemical anionic ring-opening polymerization, (b) ¹H NMR spectra of PMPG (blue), PDPG (green), PBPG (purple), PTGE (orange), and (s)-PTGE (red), and (c) the GPC traces of PMPG (blue), PDPG (green), PBPG (purple), PTGE (orange), and (s)-PTGE (red). ¹H NMR spectra were obtained in CD₂Cl₂ and CDCl₃, and GPC was conducted in THF using a RI signal with polystyrene (PS) standards. Table 1 presents a detailed characterization of the synthesized polyethers. Note that the peak intensity of PBPG is considerably lower than the others due to the limited solubility in THF.

2) *t*-BuP₄ was used in this study. In recent publications in mechanochemical polymerizations DBU and TBD, two more commonly found bases, have been found to achieve good results. The authors should elaborate on why only one base was considered in this manuscript.

We thank the reviewer for the insightful comment. In fact, we performed the ball milling polymerization initially using other bases such as DBU and TBD besides *t*-BuP₄. Unfortunately, both bases did not result in a successful polymerization possibly due to low reactivity of base (see Figure R1 below). As such, the organic superbases *t*-BuP₄ was employed throughout this study.

[Review-only material]

Figure R1. ^1H NMR spectra of ball milling polymerization by using other bases. (a) DBU; Conv. = 0.0%, and (b) TBD Conv. = 0.0% (400 MHz, CD_2Cl_2); Reaction condition: 10-mL stainless steel jar with three stainless steel balls with a diameter of 12 mm for 30 min at 30 Hz.

3) The authors claim that increasing the number of milling balls in the vessel is increasing the impact per collision. This is not correct. Due to the hindering of each other, more milling balls, in fact, lead to a reduced free path length of the milling balls and thus lower overall speed and smaller impact energy.

The higher yield, however, is an interesting phenomenon, that in my opinion is rather due to the increased number of collisions and thus a higher number of shear events leading to more reactive surfaces being created and better mixing. There has been a lot of work done by the mechanochemical community and the early works of Ondruschka and more recent works of Qwade and Boldyreva deal with this topic.

We appreciate the reviewer for this insightful comment. As commented, increasing the number of balls can decrease the mixing efficiency due to the reduced free path length of milling balls. On the contrary, it should be noted that this phenomenon is only observed in case of high filling degree of the balls in the milling jar beyond the threshold value (typically, $\sim 1/3$). Specifically, to explore mixing efficiency of ball milling reaction, the appropriate filling degree of the jar with the balls should be considered. In this context, Stolle, Schmidt, and co-workers reported that one-third is a maximum proportion of milling balls within jars for efficient mixing (see the Figure R2 below).

Our experiment condition (10-mL stainless steel jar with three 12-mm stainless steel balls) is almost one-third, which means that more balls could lead to high reactivity in presented cases

in Supplementary Table 1. Specifically, the value of filling degree in our system is 27.15%, based on the following calculation.

$$\frac{4}{3} \times \pi \times 0.6 \text{ cm}^3 = 0.905 \text{ cm}^3 \times 3 = 2.715 \text{ cm}^3 = 2.715 \text{ mL} \div 10 \text{ mL} * 100 = 27.15\%$$

A related sentence and reference are included in the revised manuscript as described below.

[Page 9 in the revised manuscript] – new sentence included is highlighted

Moreover, larger sizes and numbers of balls afforded higher conversions due to greater impact per collision. We estimated that the filling degree of the milling balls in jar is approximately 27%, which is reported to be efficient for mixing in previous study.²⁴

[Page 9 in the revised manuscript] – new reference is added

24. Schmidt, R., Burmeister, C. F., Baláž, M., Kwade, A., Stolle, A. Effect of reaction parameters on the synthesis of 5-arylidene barbituric acid derivatives in ball mills. *Org. Process Res. Dev.* **19**, 427–436 (2015)

[Review-only material]

Figure 3. Influence of Φ_{MB} on $t_{97\%}$ (a) and the stress conditions (b). Conditions: PBM P6, 250 mL steel beaker, ZrO_2 -balls, $\nu_{rot} = 650 \text{ min}^{-1}$, 100 mmol 1a, and 100 mmol 2a. $d_{MB} = 10 \text{ mm}$ in (b).

Influence of the Milling Ball Filling Degree (Φ_{MB}). As mentioned earlier, aside from the application of smaller milling balls, the stress frequency can be increased by using a larger number of milling balls. The milling ball filling degree represents the volume of the milling balls relative to the beaker volume. This important parameter influences not only the stress frequency but also the trajectories of the milling balls to affect the yield and the amount of heat dissipated by friction.^{3a,c,10,13} Fang et al. investigated the influence of Φ_{MB} on temperature in a lysis mill and observed that the temperature was maximum when 60% of the tube was filled with milling balls.¹⁴ Visualized in Figure 3a, the results showed that Φ_{MB} is an important parameter in Knoevenagel condensation in PBMs. Independent of the d_{MB} , a minimum for the milling time of approximately $0.25 \leq \Phi_{MB} \leq 0.3$ was observed in which a broader minimum could be identified for the 10 mm balls. Φ_{MB} that was either higher or lower than this range led to decreased yields, and the milling time had to be increased to reach similar conversion levels. If Φ_{MB} was less than optimal, the energy provided was reduced due to a lower number of milling balls. This led to a reduced stress frequency, and as the amount of substrate was constant, less energy was transferred to the substrate (Figure 3b). On the other hand, ball movement is hindered at high Φ_{MB} values, meaning that the milling balls have less acceleration and therefore the energy input is reduced.¹⁵ The manufacturer advises that in 250 mL

Figure R2. Effect of Reaction Parameters on the Synthesis of 5-Arylidene Barbituric Acid Derivatives in Ball Mills (Stolle and coworkers, *Org. Process Res. Dev.* **2015**, *19*, 427–436).

4) Figure S27 assignment on the structure is missing (a,b,c, etc.)

We thank the reviewer for pointing out the detailed information we missed in the manuscript. As suggested, we revised the manuscript and the Supplementary Information as below.

[Page 7 in the revised manuscript] – revised Figure 2 is included

Figure 2. Characterization of synthesized polyethers. (a) chemical structures of polyethers synthesized via mechanochemical anionic ring-opening polymerization, (b) ¹H NMR spectra of PMPG (blue), PDPG (green), **PBPG (purple)**, PTGE (orange), and (s)-PTGE (red), and (c) the GPC traces of PMPG (blue), PDPG (green), **PBPG (purple)**, PTGE (orange), and (s)-PTGE (red). ¹H NMR spectra were obtained in CD₂Cl₂ and CDCl₃, and GPC was conducted in THF using a RI signal with polystyrene (PS) standards. Table 1 presents a detailed characterization of the synthesized polyethers. **Note that the peak intensity of PBPG is considerably lower than the others due to the limited solubility in THF.**

5) The authors talk about the selection of 5 monomers and explain their reasoning well. At first, I was disappointed because no liquid monomer was used, however, in the end (line 238 ff.) they discuss the reaction with a liquid monomer. I think they should mention in their selection process, that a control experiment with a liquid monomer has been made.

We appreciate the reviewer for this thoughtful comment. As suggested, we included the description of liquid monomer in the selection process described in the introduction.

[Page 6 in the revised manuscript] – new sentence included is highlighted

The five representative solid-state monomers of MPG (R1), DPG (R2), BPG (R3), TGE (R4), and (*s*)-TGE (R5) were selected from 13 candidates on the basis of their phase and solubility (Supplementary Fig. 1). Additionally, a liquid monomer, benzyl glycidyl ether, was selected for the comparison between solid- and solution-state mechanochemical polymerization. Further details on the ball milling polymerization of liquid monomer is described in the following section.

6) Suggestions for further work:

In line 213ff the authors discuss the fact, that they cannot control the first reaction regime, because an initial mixing during the preparation of the milling jar is unavoidable. Several mechanochemists have had the same problem and in some of their works, Hernandez and Bolm used glass capillaries to seal one of their reactants which only breaks when the milling is initialized. Especially for the field of controlled polymerization of soluble polymers, this technique could be highly beneficial.

We appreciate reviewer for insightful comment. As suggested by the reviewer and also practiced by the other leading researchers in the field, we performed the mechanochemical polymerization of some of monomers using a glass capillary. Specifically, three monomers such as MPG, DPG, and BPG were tested using a glass capillary filled with benzyl alcohol initiator and *t*-BuP₄ base to exclude unavoidable initial mixing issues. Interestingly, we found that almost identical polymerization results were observed even in this case, which confirms that this technique could not bypass the initial mixing issues found in this study (see Figure R3, and R4 below). We also found that the glass capillary broken within the jar serves as grinding auxiliary to facilitate the initial polymerization reactivity.

[Review-only material]

Figure R3. (a) Photographs of glass capillaries used to seal the initiator complex from monomer and (b) ball-milling reaction using prepared glass capillaries.

Figure R4. Ball-milling polymerization of (a) MPG, (b) DPG, and (c) BPG using glass capillaries. Note that the glass capillary broken within the jar serves as grinding auxiliary to facilitate the initial polymerization reactivity.

Reviewer: 3

Comments:

The authors present an anionic ring-opening polymerization of functional epoxide monomers in the solid state using a mechanochemistry approach. The authors synthesized polyethers from diverse functional epoxide monomers using solid-state and solution polymerization reactions. To understand the distinctive reactivity of ball milling, the key characteristics of the monomers were examined. The reaction conditions were optimized with different types of jars and balls as well as different times. The results showed bulky monomers exhibited faster conversions in the solid-state contrast to that observed for solution polymerization. This was confirmed by NMR, GPC, and MALDI-ToF analyses. The authors showed linear correlation was observed between the conversion of the ball milling polymerization and the melting point of the functional epoxide monomers, indicating the melting point is a significant factor of mechanochemical polymerization reactivity. **Overall, this is a well-written and organized manuscript highlighting a better understanding of ring-opening polymerization synthesis using the mechanochemistry approach. So, this manuscript is worthy of publication in the Nature Communication Journal with revisions.**

1) In Table 1 and Supplementary Table 1, I would like to see DP from the ^1H NMR spectrum for each isolated polymer. In addition, it's worth adding the Initiator efficiency (IE%) of the BnOH from the crude ^1H NMR spectrum for each polymer which has been obtained by different researchers such as Nelson and coworkers, *Macromolecules* 2022, 55, 21, 9740.

We appreciate the reviewer for comment. As suggested, we added the DP values for all polymers using the ^1H NMR analysis and included these DP values in Table 1 in the revised manuscript and Supplementary Table 1 in the revised supplementary information.

Meanwhile, Nelson and coworkers calculated the initiator efficiency (IE%) considering the reacted initiator and unreacted initiator. In our system, however, it is hard to distinguish the chemical shift values of each reacted and unreacted initiators, thus limiting the determination of IE% by using ^1H NMR. Alternatively, we employed MALDI-ToF analysis for calculation of IE% for some representative polymers such as of PTGE and of (*s*)-PTGE – those proceeding well under mechanochemical polymerizations.

As confirmed in Supplementary Fig. 42, the IE% of (*s*)-PTGE using BnOH was determined to be approximately 50%. While this value is not as high as we expected, notably, this could be the minimum IE% considering the potential degradation occurring in BnOH under MALDI measurement conditions. The other fraction was originated from the self-initiation which is inevitably observed in mechanochemical polymerization proceeding under ambient conditions.

[Page 8 in the revised manuscript] – new data included in Table 1 is highlighted

Table 1. Characterization of the synthesized polyethers via ball milling.

Polymer	Conv. ^a (%)	$M_{n,th}$ (g/mol)	$M_{n,NMR}$ (g/mol)	DP_{NMR}	$M_{n,GPC}^d$ (g/mol)	\bar{D}^d	T_g^e (°C)	mp ^f (°C)
PMPG	47.7 ± 4.8	6470	31790 ^b	176	2380	1.21	9.0	47.8
PDPG	64.1 ± 12.5	8050	8760 ^b	41	2760	1.17	13.1	63.8
PBPG	86.5 ± 0.2	9880	n.d.	n.d.	2100	1.12	204.4 ^g	90.6
PTGE	90.8 ± 0.2	14080	23490 ^c	74	3720	1.13	74.0	87.8
(s)-PTGE	92.1 ± 1.0	14420	19180 ^c	60	5950	1.15	82.0	101.2

All polymerizations were targeted to a DP of 50 and conducted for 30 min. ^aMonomer conversion as calculated from the ¹H NMR spectrum of the crude monomer ($n = 3$, average values reported with standard deviation). ^bCalculated from the ¹H NMR spectrum of the isolated polymer (CDCl₃, 400 MHz). ^cCalculated from the ¹H NMR spectrum of the isolated polymer (CD₂Cl₂, 400 MHz) ^dMeasured via GPC in THF using PS standard and RI signal. ^eDetermined via DSC at a rate of 10 °C/min. ^fMelting point of the epoxide monomer measured via DSC. ^g T_m .

[Page S-40 in the revised Supporting Information] – new data included in Supplementary Table 1 is highlighted

Supplementary Table 1. Investigation of ball milling parameters. (a) Effect of type of jar and balls, (b) effect of size of balls, and (c) effect of the number of balls. All polymerization was performed using MPG monomer.

(a)

Entry	Type of jar & balls	Size of balls (mm)	# of balls	Time (h)	Conv. (%)	DP_{NMR}
1	Teflon & ZrO ₂	5	4	6	33	8
2	Stainless & Stainless	7	3	6	>99	100

(b)

Entry	Type of jar & balls	Size of balls (mm)	# of balls	Time (h)	Conv. (%)	DP_{NMR}
1	Stainless & Stainless	7	3	0.5	20	n.d.
2	Stainless & Stainless	12	3	0.5	48	176

(c)

Entry	Type of jar & balls	Size of balls (mm)	# of balls	Time (h)	Conv. (%)	DP_{NMR}
1	Stainless & Stainless	7	3	6	>99	100
2	Stainless & Stainless	7	10	4	>99	130
3	Stainless & Stainless	7	20	2	>99	366

2) On page 6, line 102, Table 1 should be placed directly after the end of this paragraph.

We thank the reviewer for comment. As suggested, Table 1 was located after the end of the paragraph in the revised manuscript.

[Page 8 in the revised manuscript] – Position of Table 1 is changed

This result clearly supports the absence of racemization during ball milling polymerization, which is often observed in solution polymerization.

Table 1. Characterization of the synthesized polyethers via ball milling.

Polymer	Conv. ^a (%)	$M_{n,th}$ (g/mol)	$M_{n,NMR}$ (g/mol)	DP_{NMR}	$M_{n,GPC}^d$ (g/mol)	\bar{D}^d	T_g^e (°C)	mp ^f (°C)
PMPG	47.7 ± 4.8	6470	31790 ^b	176	2380	1.21	9.0	47.8
PDPG	64.1 ± 12.5	8050	8760 ^b	41	2760	1.17	13.1	63.8
PBPG	86.5 ± 0.2	9880	n.d.	n.d.	2100	1.12	204.4 ^g	90.6

PTGE	90.8 ± 0.2	14080	23490 ^c	74	3720	1.13	74.0	87.8
(s)-PTGE	92.1 ± 1.0	14420	19180 ^c	60	5950	1.15	82.0	101.2

All polymerizations were targeted to a DP of 50 and conducted for 30 min. ^aMonomer conversion as calculated from the ¹H NMR spectrum of the crude monomer ($n = 3$, average values reported with standard deviation). ^bCalculated from the ¹H NMR spectrum of the isolated polymer (CDCl₃, 400 MHz). ^cCalculated from the ¹H NMR spectrum of the isolated polymer (CD₂Cl₂, 400 MHz) ^dMeasured via GPC in THF using PS standard and RI signal. ^eDetermined via DSC at a rate of 10 °C/min. ^fMelting point of the epoxide monomer measured via DSC. ^g T_m .

The effects of various ball milling parameters, including the materials of the jar and balls, size of the balls, and number of balls, on mechanochemical AROP of the functional monomers were investigated (Supplementary Table 1).

3) On page 7, there is no data provided for the control reactions such as M_n , DP, Conv% and IE%, so, for a clear comparison with results from Table 1, I would like to see a table for these reactions.

We appreciate the reviewer for comment. As suggested, we updated the Table in the revised Supplementary Information for all monomers that were polymerized in solution-state for comparison (see Table R1). Once again, we could find that there is a clear opposite trend in the reactivity order of the monomers depending on the mode of polymerizations (solution- vs solid-state mechanochemical polymerization). Although a direct comparison is difficult as the conversion of monomer varies depending on the mode of polymerization, we could observe that the reactivity tendency is preserved.

Furthermore, IE% of the representative (s)-PTGE polymer was calculated via MALDI-ToF analysis (see Figure R5, and R6 below). It was found that IE% of BnOH initiator in the solution-state polymerization was approximately 65%, which is higher than mechanochemical reaction under ambient condition (i.e., ~50%). It is of note that the solid-state mechanochemical polymerization can induce the inevitable self-initiation of monomers considering a high concentration of monomers present.

[Review-only material]

Table R1. Characterization of the synthesized polyethers via solution-state.

Polymer	Conv. ^a (%)	$M_{n,th}$ (g/mol)	$M_{n,NMR}$ (g/mol)	DP_{NMR}	$M_{n,GPC}^d$ (g/mol)	D^d
PMPG	92.5	8680	12350 ^b	73	710	1.60
PDPG	>99	10610	8720	41	3110	1.18
PBPG	91.2	10410	n.d.	n.d.	1870	1.18
PTGE	75.0	11960	11800 ^c	43	1870	1.16
(s)-PTGE	69.7	11120	11490 ^c	36	2970	1.16

All polymerizations were targeted to a DP of 50 and conducted for 30 min. ^aMonomer conversion as calculated from the ¹H NMR spectrum of the crude monomer. ^bCalculated from the ¹H NMR spectrum of the isolated polymer (CDCl₃, 400 MHz). ^cCalculated from the ¹H NMR spectrum of the isolated polymer (CD₂Cl₂, 400 MHz) ^dMeasured via GPC in THF using PMMA standard and RI signal.

[Review-only material]

Figure R5. (a) MALDI-ToF MS spectrum of the isolated (*s*)-PTGE polyether via solution polymerization with individual peak assignments in the selected region. Experimental conditions: reflector positive mode, and 2,5-dihydroxybenzoic acid matrix.

Figure R6. Distribution histogram of individual initiating group in the resulting (*s*)-PTGE polyether displayed in Figure R5.

4) In Table 1, the data shows that $M_{n,NMR}$ values for all polymers were much higher than $M_{n,GPC}$. So, I am wondering about the reason for this.

We appreciate the reviewer for this critical comment. The main reason for different $M_{n,NMR}$ with $M_{n,GPC}$ is a standard materials of GPC. Polystyrene is used for standard material in all synthesized polymer because the phenyl group is contained in all prepared polymers. Accordingly, a deviation of hydrodynamic radius in GPC solvent would be resulted in, which

induces the difference of molecular weight from GPC and NMR analysis. Moreover, a significantly high hydrophobicity of the phenyl, biphenyl and trityl groups used in this study could decrease the effective hydrodynamic radius of polymers in GPC solvent (see Figure R7). A similar phenomenon was observed in our previous study using a monomer with considerably high hydrophobicity.

[Review-only material]

(Figure S8). It is worth noting that the $M_{n, \text{GPC}}$ values tend to be considerably lower than the $M_{n, \text{NMR}}$ values, as shown in Table 1. This observation could be attributed to the high hydrophobicity of the CHGE monomer in the homopolymers and block copolymers, which in turn led to the difference in their hydrodynamic volumes in the solvent. With increasing the fraction of the CHGE block, this difference in their molecular weights becomes more pronounced as indicated in both homopolymers and block copolymers. To confirm the

Table 1. Characterization Data for All Polymers Synthesized in This Study

polymer code	polymer composition ^a	$M_{n, \text{NMR}}^b$ (g mol ⁻¹)	$M_{n, \text{GPC}}^c$ (g mol ⁻¹)
	PCHGE ₂₅	5100	4000 ^c
	PCHGE ₅₀	10,100	2400 ^c
CH7	mPEG ₁₄₄ -b-PCHGE ₇	6400	3900 ^d
CH28	mPEG ₁₄₄ -b-PCHGE ₂₈	10,600	3500 ^d
CH45	mPEG ₁₄₄ -b-PCHGE ₄₅	14,000	2700 ^d
E60	mPEG ₁₄₄ -b-PCHGE ₆₀	14,700	14,700
P37	mPEG ₁₄₄ -b-PCHGE ₃₇	11,800	11,600
F47	mPEG ₁₄₄ -b-PCHGE ₄₇	8900	15,300

^aPolymerization conversion was over 99%. ^bDetermined via ¹H NMR spectroscopy. ^cMeasured by GPC (THF, RI signal, PMMA standard). ^dMeasured by GPC (DMF, RI signal, PMMA standard). ^e T_g was determined by the differential scanning calorimetry (DSC) at a rate of 10 °C min⁻¹. ^fThe CMC value was calculated by fluorescence spectroscopy with pyrene as the probe. Polymers E, P, and F have been described in previous studies.^{29,30}

Figure R7. Representative example displaying the molecular weight difference between $M_{n, \text{NMR}}$ and $M_{n, \text{GPC}}$ in the case of high hydrophobic monomers. (Kim and coworkers, “pH-Responsive Amphiphilic Polyether Micelles with Superior Stability for Smart Drug Delivery” *Biomacromolecules* **2021**, *22*, 2043–2056).

5) The authors discuss conversion %, $M_{n, \text{th}}$, and $M_{n, \text{NMR}}$, but never define it or explain how they calculated it. Please define these parameters in addition to the DP_{NMR} as well as IE% by using the equation and explaining how they calculated it.

We thank reviewer for careful comment. It is explained that how to calculate the conversion and $M_{n,NMR}$ in last paragraph of page 8. As suggested, we presented the equations for determining each conversion, $M_{n,th}$, $M_{n,NMR}$, DP_{NMR} , and IE% as follow (see Figure R8–R12).

[Review-only material]

(a)

$$\text{Conversion} = \frac{\int \text{aromatic H in polymer}}{\{\int \text{aromatic H in polymer} + (\int \text{methylene H (a) in unreacted monomer} * \# \text{ of aromatic H})\}} * 100(\%)$$

(b)

$$\text{Conversion} = \frac{\int \text{methylene backbone H in polymer}}{[\int \text{methylene backbone H in polymer} + (\int \text{methylene H (a) in unreacted monomer} * 2)]} * 100(\%)$$

Figure R8. Equation for mechanochemical polymerization conversion of (a) PMPG and PDPD, (b) PTGE and (s)-PTGE.

$$M_{n,th} = \text{molecular weight of initiator} + (\text{molecular weight of monomer} \times \text{target DP})$$

Figure R9. Equation for theoretical molecular weight of polymer.

(a)

$$M_{NMR} = \left(\frac{\int \text{aromatic proton in polymer}}{\# \text{ of aromatic protons}} * \text{molecular weight of monomer} \right) + \text{molecular weight of initiator}$$

(b)

$$M_{NMR} = \left(\frac{\int \text{methylene proton in polymer}}{\# \text{ of methylene protons}} * \text{molecular weight of monomer} \right) + \text{molecular weight of initiator}$$

Figure R10. Equation for molecular weight of (a) PMPG and PDPG, (b) PTGE and (s)-PTGE calculated from ^1H NMR spectrum.

(a)

$$DP_{NMR} = \int (\text{aromatic proton in polymer} / \# \text{ of aromatic protons})$$

(b)

$$DP_{NMR} = \int (\text{methylene proton in polymer} / \# \text{ of methylene protons})$$

Figure R11. Equation for degree of polymerization (DP) of (a) PMPG and PDPG, (b) PTGE and (s)-PTGE calculated from ¹H NMR spectrum.

(a)

$$IE\% = \frac{\int \textit{initiation from BnOH initiator}}{(\int \textit{initiation from BnOH initiator} + \int \textit{self - initiation})} * 100$$

(b)

$$IE\% = \frac{\int \textit{self - initiation}}{(\int \textit{initiation from BnOH initiator} + \int \textit{self - initiation})} * 100$$

Figure R12. Equation for initiation efficiency (IE%) calculated from MALDI-ToF spectrum at specific degree of polymerization (a) BnOH initiation and (b) self-initiation.

6) In the introduction, by the end of line 30, a reference for this paragraph should be added.

We appreciate the reviewer for helpful comment. As suggested, we included a related reference in the revised manuscript as below.

[Page 3 in the revised manuscript] – new reference added is highlighted

To understand and control mechanochemical reactions, particularly in the case of ball milling reactions, various parameters influencing the reactivity have been investigated. These parameters typically include the type of milling materials, size and number of balls, period and frequency of milling, grinding auxiliary, and liquid used to assist grinding.⁴

4. Krusenbaum, A., Grätz, S., Tigineh, G.T., Borchardt, L., Kim, J.G. The mechanochemical synthesis of polymers. *Chem. Soc. Rev.* **51**, 2873–2905 (2022).

7) In Supplementary Information, a Table of content needs to be added.

We appreciate the reviewer for comment. As suggested, we included a Table of content in Supplementary Information.

Reviewers' Comments:

Reviewer #1:

Remarks to the Author:

Although this revision shows some improvement in controlling the reaction temperature, which was a critical problem in the previous version, the fundamental flaw in this paper does not seem to have been improved. This paper is about the rate difference of monomers, but in the first place, the rate difference between systems with different reaction temperatures cannot be discussed quantitatively. While previous results could not rule out an increase in reactivity such as PTGE simply due to an increase in temperature caused by friction or reaction heat, the use of a cooling jacket qualitatively shows the reversal of the reactivity trend. However, the data still do not allow a quantitative discussion, and it is highly inappropriate to make a quantitative argument, for example, as in Figures 4b-d. Also, it is not possible to explain the differences in reactivity at the molecular level, as previously requested. In the response, the author commented, "We used "rigidity" as a collective term to represent high physical and chemical properties of monomers, including melting point, molecular weight, and possibly all other related inter- and intramolecular interactions," but I do not see how such multiple different concepts can be combined into one word. The ambiguity of this paper is also due to the fact that it uses several different concepts in a single word.

1. The reaction temperature (or the temperature at which the jar was emptied) should be listed in Table 1
2. Please discuss why the reactivity reversal occurs at the molecular level, instead of using the undefined word "rigidity" as a cover for it.
3. Figure 4b-d should be deleted.

Reviewer #2:

Remarks to the Author:

The authors have made significant improvements to the manuscript by addressing all the points raised by the referees. The additional experiments, especially the ones with thermal control, conducted provide strong support for the points raised and significantly enhance the overall manuscript.

However, I still have a concern regarding Figure S27. It is crucial that the authors ensure consistent labeling of the protons in Figure S27, aligning it with Figure 2 in the main manuscript. This consistency will enhance the clarity and cohesiveness of the figures presented. (See sketch below)

Figure 2. Characterization of synthesized polyethers. (a) chemical structures of polyethers

Furthermore, I strongly recommend that point 5, as raised by reviewer 3, along with the authors' response, be included in the Electronic Supporting Information (ESI) rather than being limited to the "Review-only material" section. While the equations discussed may be considered common knowledge, including this information in the ESI would be immensely helpful for researchers who are new to the field and embarking on their research journey. It would contribute to the overall educational value of the manuscript and aid in disseminating knowledge to a broader audience.

Reviewer #3:

Remarks to the Author:

The authors have appropriately responded to all my concerns and comments so I approve this manuscript for publication in its current state.

Reviewer(s)' Comments to Author:

Reviewer: 1

Comments:

Although this revision shows some improvement in controlling the reaction temperature, which was a critical problem in the previous version, the fundamental flaw in this paper does not seem to have been improved. This paper is about the rate difference of monomers, but in the first place, the rate difference between systems with different reaction temperatures cannot be discussed quantitatively. While previous results could not rule out an increase in reactivity such as PTGE simply due to an increase in temperature caused by friction or reaction heat, the use of a cooling jacket qualitatively shows the reversal of the reactivity trend. However, the data still do not allow a quantitative discussion, and it is highly inappropriate to make a quantitative argument, for example, as in Figures 4b-d. Also, it is not possible to explain the differences in reactivity at the molecular level, as previously requested. In the response, the author commented, "We used "rigidity" as a collective term to represent high physical and chemical properties of monomers, including melting point, molecular weight, and possibly all other related inter- and intramolecular interactions," but I do not see how such multiple different concepts can be combined into one word. The ambiguity of this paper is also due to the fact that it uses several different concepts in a single word.

We appreciate the reviewer for this critical comment. We understand that the reviewer raised the importance of controlling the temperature during the mechanochemical polymerization under ball milling to make the discussion on the rate difference between systems valid. Although it is very challenging to keep a constant temperature during mechanochemical reactions, it should be noted that mechanical force, not temperature, is the main energy source driving the polymerization in this system, thereby exhibiting the unique reaction trend compared to the conventional solution-based method.

While the quantitative comparison may not be entirely acceptable in this case as indicated by the reviewer, most importantly, we believe that proposing a semiempirical relationship in deciphering and/or predicting the reactivity of the monomers during the mechanochemical polymerization by itself deserves its own merit, as the field of mechanochemical polymerization is still in its infancy and rapidly growing field. This is also acknowledged by the reviewer 2 in our first revision, "The paper is of significant interest to the polymer community as a whole and the mechanochemical community in particular".

As such, we sincerely request the reviewer to understand the current system could provide a room for more dynamic discussion on the topic of mechanochemical polymerization. Nonetheless, we included the following sentence to highlight that this comparison is only valid

to represent the difference in reactivity among different monomers tested in this study (see below).

[Page 15 in the revised manuscript] – new sentences included are highlighted

Surprisingly, as mechanochemical polymerization progresses, close linear correlations were found between the conversion and both the melting points and molecular weights of the monomers. Although it is challenging to decouple the physical properties of monomers individually, this result suggests that the melting point of monomers, which integrates various intermolecular interactions, is the most critical predictor of ball milling polymerization reactivity. It can be proposed that the reactivity reversal observed in the solid-state polymerization is possibly originated from that the bulky substituents with a higher intermolecular interaction in the monomer (e.g. melting point) can recruit more mechanochemical forces during the ball-milling polymerization. On the other hand, the bulky substituent could cause retardation of the polymerization in solution due to steric hindrance. However, it should be considered that quantitative analysis of the reactivity is only valid when the reaction temperature is constant during mechanochemical reactions of different monomers (Supplementary Fig. 60). Nonetheless, we believe that proposing a semiempirical relationship in deciphering and/or predicting the reactivity of the monomers during the mechanochemical polymerization by itself deserves its own merit, as the field of mechanochemical polymerization is still in its infancy and rapidly growing field.

1) The reaction temperature (or the temperature at which the jar was emptied) should be listed in Table 1.

We appreciate the reviewer for the comment. As suggested, we added the temperature of the jar (T_{jar}) in Table 1 of the revised manuscript.

[Page 8 in the revised manuscript] – new data included in Table 1 are highlighted

Table 1. Characterization of the synthesized polyethers via ball milling.

Polymer	Conv. ^a (%)	$M_{n,\text{th}}$ (g/mol)	$M_{n,\text{NMR}}$ (g/mol)	DP_{NMR}	$M_{n,\text{GPC}}^d$ (g/mol)	\bar{D}^d	T_g^e (°C)	mp ^f (°C)	T_{jar}^h (°C)
PMPG	47.7 ± 4.8	6470	31790 ^b	176	2380	1.21	9.0	47.8	41.2

PDPG	64.1 ± 12.5	8050	8760 ^b	41	2760	1.17	13.1	63.8	44.5
PBPG	86.5 ± 0.2	9880	n.d.	n.d.	2100	1.12	204.4 ^g	90.6	61.3
PTGE	90.8 ± 0.2	14080	23490 ^c	74	3720	1.13	74.0	87.8	56.8
(s)-PTGE	92.1 ± 1.0	14420	19180 ^c	60	5950	1.15	82.0	101.2	50.0

All polymerizations were targeted to a DP of 50 and conducted for 30 min. ^aMonomer conversion as calculated from the ¹H NMR spectrum of the crude monomer ($n = 3$, average values reported with standard deviation). ^bCalculated from the ¹H NMR spectrum of the isolated polymer (CDCl₃, 400 MHz). ^cCalculated from the ¹H NMR spectrum of the isolated polymer (CD₂Cl₂, 400 MHz) ^dMeasured via GPC in THF using PS standard and RI signal. ^eDetermined via DSC at a rate of 10 °C/min. ^fMelting point of the epoxide monomer measured via DSC. ^gT_m. ^hTemperature inside the jar after ball milling reaction measured by IR thermometer.

2) Please discuss why the reactivity reversal occurs at the molecular level, instead of using the undefined word "rigidity" as a cover for it.

We appreciate the reviewer for this critical comment. In fact, we do not have clear evidence at the moment for this because this reactivity reversal is first observed in this mechanochemical polymerization of functional epoxide monomers. It can be proposed that the reactivity reversal observed in the solid-state polymerization is possibly originated from that the bulky substituents with a higher intermolecular interaction in the monomer (e.g. melting point) can recruit more mechanochemical forces during the solid-state mechanochemical polymerization. Thereby, this high mechanical force can be utilized to drive the polymerization in the solid-state. On the other hand, the bulky substituent could prevent the polymerization in solution by hindering the approach of incoming monomer to the active chain end. This discussion is briefly included in the revised manuscript as below.

[Page 15 in the revised manuscript] – new sentences included in revised manuscript are highlighted

Surprisingly, as mechanochemical polymerization progresses, close linear correlations were found between the conversion and both the melting points and molecular weights of the monomers. Although it is challenging to decouple the physical properties of monomers individually, this result suggests that the melting point of monomers, which integrates various

intermolecular interactions, is the most critical predictor of ball milling polymerization reactivity. It can be proposed that the reactivity reversal observed in the solid-state polymerization is possibly originated from that the bulky substituents with a higher intermolecular interaction in the monomer (e.g. melting point) can recruit more mechanochemical forces during the ball-milling polymerization. On the other hand, the bulky substituent could cause retardation of the polymerization in solution due to steric hindrance.

Once again, it is true indeed that it is very difficult to combine the multiple physical properties into a single word such as rigidity. To reflect the comment from the reviewer, we removed the term “rigidity” in the revised manuscript and modified the title of the revised manuscript.

[Page 1 in the revised manuscript, and Supplementary Information] – Revised title is highlighted

Functional Epoxide Monomers in the Solid State: How Does **the Molecular Structure** Influence the Polymerization?

[Page 15 in the revised manuscript] – Revised word included is highlighted

Inspired by the unique reactivity during mechanochemical polymerization, we were prompted to correlate the reactivity with the physical properties of the monomers. Figure 4a depicts the effect of monomer **molecular structure** on its mechanochemical polymerization reactivity. Moreover, as the **molecular structure** of the monomers could be deduced from their melting points and molecular weights, the corresponding reactivity trends during ball milling polymerization could be explored by plotting the conversion against melting point (Figure 4b–d) and against molecular weight (Supplementary Fig. 61) at various reaction times.

3) Figure 4b-d should be deleted.

We appreciate the reviewer for comment. As addressed in the previous responses, we do not concur with the suggestion by the reviewer. While the quantitative comparison may not be entirely acceptable in this case as indicated by the reviewer, most importantly, we believe that proposing a semiempirical relationship in deciphering and/or predicting the reactivity of the monomers during the mechanochemical polymerization by itself deserves its own merit, as the field of mechanochemical polymerization is still in its infancy and rapidly growing field.

To clarify the concern raised by the reviewer, we included the following sentences in the revised manuscript.

[Page 15 in the revised manuscript] – new sentences included are highlighted

Surprisingly, as mechanochemical polymerization progresses, close linear correlations were found between the conversion and both the melting points and molecular weights of the monomers. Although it is challenging to decouple the physical properties of monomers individually, this result suggests that the melting point of monomers, which integrates various intermolecular interactions, is the most critical predictor of ball milling polymerization reactivity. It can be proposed that the reactivity reversal observed in the solid-state polymerization is possibly originated from that the bulky substituents with a higher intermolecular interaction in the monomer (e.g. melting point) can recruit more mechanochemical forces during the ball-milling polymerization. On the other hand, the bulky substituent could cause retardation of the polymerization in solution due to steric hindrance. However, it should be considered that quantitative analysis of the reactivity is only valid when the reaction temperature is constant during mechanochemical reactions of different monomers (Supplementary Fig. 60). Nonetheless, we believe that proposing a semiempirical relationship in deciphering and/or predicting the reactivity of the monomers during the mechanochemical polymerization by itself deserves its own merit, as the field of mechanochemical polymerization is still in its infancy and rapidly growing field.

Reviewer #2 (Remarks to the Author):

The authors have made significant improvements to the manuscript by addressing all the points raised by the referees. The additional experiments, especially the ones with thermal control, conducted provide strong support for the points raised and significantly enhance the overall manuscript.

1) However, I still have a concern regarding Figure S27. It is crucial that the authors ensure consistent labeling of the protons in Figure S27, aligning it with Figure 2 in the main manuscript. This consistency will enhance the clarity and cohesiveness of the figures presented.

We appreciate the reviewer for this careful comment we missed in the manuscript. As suggested, we revised the Supplementary Information as below.

[Page S-32 in the revised Supplementary Information] – Revised Supplementary Fig. S27 is included

Supplementary Fig. 27. ¹H NMR spectrum of (s)-PTGE polymer obtained via ball milling AROP for 2 h (400 MHz, CD₂Cl₂): Conv. = 94.4%.

2) Furthermore, I strongly recommend that point 5, as raised by reviewer 3, along with the authors' response, be included in the Electronic Supporting Information (ESI) rather than being limited to the "Review-only material" section. While the equations discussed may be considered common knowledge, including this information in the ESI would be immensely helpful for researchers who are new to the field and embarking on their research journey. It would contribute to the overall educational value of the manuscript and aid in disseminating knowledge to a broader audience.

We appreciate reviewer for the comment. As suggested, we included these equations in the Supplementary Information as below.

[Page 7 in the revised manuscript] – Revised equations included are highlighted

Further, the DP and corresponding number-average molecular weights ($M_{n,NMR}$) of resulting polyethers were calculated from the ratio of the protons of benzyl alcohol initiator (x) at 4.48 ppm to the polymeric protons at 4.21–3.45 (PMPG), 4.09–3.45 (PDPG), 3.22–2.92 (PTGE), 3.20–2.91 ppm ((*s*)-PTGE), respectively (Table 1, and **Supplementary Eq. 1–4**).

[Page 10 in the revised manuscript] – Revised equation included is highlighted

Specifically, a higher fraction of the peaks from the initiation by benzyl alcohol was observed in the polymerization of (*s*)-PTGE compared with self-initiation (Supplementary Fig. 42, and **Supplementary Eq. 5**).

[Page S-69 in the revised Supplementary Information] – New Supplementary Eq. 1–5 are included

(a)

$$\text{Conversion} = \frac{\int \text{aromatic H in polymer}}{\int \text{aromatic H in polymer} + \left\{ \int \text{methylene H (a) in unreacted monomer} \times \# \text{ of aromatic H} \right\}} \times 100 (\%)$$

(b)

$$\text{Conversion} = \frac{\int \text{methylene backbone H in polymer}}{\int \text{methylene backbone H in polymer} + \left\{ \int \text{methylene H (a) in unreacted monomer} \times 2 \right\}} \times 100 (\%)$$

Supplementary Eq. 1. Mechanochemical polymerization conversion of (a) PMPG and PDPD, (b) PTGE and (s)-PTGE.

$$M_{n,th} = \text{Molecular weight of initiator} + (\text{molecular weight of monomer} \times \text{target DP})$$

Supplementary Eq. 2. Theoretical molecular weight of polymer.

(a)

$$M_{n,NMR} = \left(\frac{\int \text{aromatic proton in polymer}}{\# \text{ of aromatic protons}} \times \text{Molecular weight of monomer} \right) + \text{molecular weight of initiator}$$

(b)

$$M_{n,NMR} = \left(\frac{\int \text{methylene proton in polymer}}{\# \text{ of methylene protons}} \times \text{Molecular weight of monomer} \right) + \text{molecular weight of initiator}$$

Supplementary Eq. 3. Molecular weight of (a) PMPG and PDPG, (b) PTGE and (s)-PTGE calculated from ¹H NMR spectrum.

(a)

$$DP_{NMR} = \int (\text{aromatic proton in polymer} / \# \text{ of aromatic protons})$$

(b)

$$DP_{NMR} = \int (\text{methylene proton in polymer} / \# \text{ of methylene protons})$$

Supplementary Eq. 4. Degree of polymerization (DP) of (a) PMPG and PDPG, (b) PTGE and (s)-PTGE calculated from ¹H NMR spectrum.

(a)

$$\text{IE\%} = \frac{\int \text{initiation from BnOH initiator}}{\int \text{initiation from BnOH initiator} + \int \text{self initiation}} \times 100 (\%)$$

(b)

$$\text{IE\%} = \frac{\int \text{self initiation}}{\int \text{initiation from BnOH initiator} + \int \text{self initiation}} \times 100 (\%)$$

Supplementary Eq. 5. Initiation efficiency (IE%) calculated from MALDI-ToF spectrum at specific degree of polymerization (a) BnOH initiation and (b) self-initiation.

Reviewer #3 (Remarks to the Author):

The authors have appropriately responded to all my concerns and comments so I approve this manuscript for publication in its current state.

We truly appreciate the reviewer for this valuable comment.

Reviewers' Comments:

Reviewer #1:

Remarks to the Author:

The results of this study are themselves very interesting and worthy of publication as a paper. My point was that from a scientific point of view, the treatment and presentation of the results is incorrect. This has nothing to do with the author's claim that the field of mechanochemical polymerization is immature or not. The author acknowledges that the reaction rate cannot be discussed quantitatively because the temperature cannot be kept constant. However, Figure 4b-d is contradictory because it clearly intends a quantitative discussion. This should be changed to more qualitative figures. For example, the table should be replaced with a table that uses terms such as "large", "medium", or "small".

Other items appear to be appropriately revised.

Reviewer(s)' Comments to Author:

Reviewer: 1

The results of this study are themselves very interesting and worthy of publication as a paper. My point was that from a scientific point of view, the treatment and presentation of the results is incorrect. This has nothing to do with the author's claim that the field of mechanochemical polymerization is immature or not. The author acknowledges that the reaction rate cannot be discussed quantitatively because the temperature cannot be kept constant. However, Figure 4b-d is contradictory because it clearly intends a quantitative discussion. This should be changed to more qualitative figures. For example, the table should be replaced with a table that uses terms such as "large", "medium", or "small".

We appreciate the reviewer for this critical comment again. To reflect the final suggestion from the reviewer, we modified the Figure 4 as below to highlight the qualitative discussion of the mechanochemical reactivity with respect to the molecular structure. The original discussion was moved to the Supplementary Fig. 61.

[Page 14 in the revised manuscript] – Revised Figure 4 is included with new caption

Figure 4. Schematic illustration of the mechanochemical polymerization reactivity of functional monomers with respect to their molecular structures.

[Page 14 in the revised manuscript] – new sentence included in revised manuscript is highlighted

Inspired by the unique reactivity during mechanochemical polymerization, we were prompted to correlate the reactivity with the physical properties of the monomers. Figure 4 depicts the effect of monomer structure on its mechanochemical reactivity. Moreover, as the intermolecular interactions of the monomers could be deduced from their melting points and molecular weights, the corresponding reactivity trends during ball milling polymerization could be explored by plotting the conversion against melting point and against molecular weight at various reaction times (Supplementary Fig. 61).

[Page S-67 in the revised revised Supplementary Information] – Revised Supplementary Fig. 61

Supplementary Fig. 61. Series of plots of monomer conversion vs. melting point or molecular weight for the various functional epoxide monomers. (a–c) Series of plots of monomer conversion vs. melting point for various functional epoxide monomers at reaction times of (a) 10 min, (b) 20 min, and (c) 30 min, and (d–f) series of plots of monomer conversion vs. molecular weight for the various functional epoxide monomers at reaction times of (d) 10 min, (e) 20 min, and (f) 30 min. All data were collected in triplicate, and the average values were reported with standard deviation.